# Policy Mirror Descent with Lookahead

Kimon Protopapas[*]    Anas Barakat[*]

## Abstract

Policy Mirror Descent (PMD) stands as a versatile algorithmic framework encompassing several seminal policy gradient algorithms such as natural policy gradient, with connections with state-of-the-art reinforcement learning (RL) algorithms such as TRPO and PPO. PMD can be seen as a soft Policy Iteration algorithm implementing regularized 1-step greedy policy improvement. However, 1-step greedy policies might not be the best choice and recent remarkable empirical successes in RL such as AlphaGo and AlphaZero have demonstrated that greedy approaches with respect to multiple steps outperform their 1-step counterpart. In this work, we propose a new class of PMD algorithms called $h$-PMD which incorporates multi-step greedy policy improvement with lookahead depth $h$ to the PMD update rule. To solve discounted infinite horizon Markov Decision Processes with discount factor $\gamma$, we show that $h$-PMD which generalizes the standard PMD enjoys a faster dimension-free $\gamma^h$-linear convergence rate, contingent on the computation of multi-step greedy policies. We propose an inexact version of $h$-PMD where lookahead action values are estimated. Under a generative model, we establish a sample complexity for $h$-PMD which improves over prior work. Finally, we extend our result to linear function approximation to scale to large state spaces. Under suitable assumptions, our sample complexity only involves dependence on the dimension of the feature map space instead of the state space size.

## 1   Introduction

Policy Mirror Descent (PMD) is a general class of algorithms for solving reinforcement learning (RL) problems. Motivated by the surge of interest in understanding popular Policy Gradient (PG) methods, PMD has been recently investigated in a line of works [25, 53, 29, 20]. Notably, PMD encompasses several PG methods as particular cases via its flexible mirror map. A prominent example is the celebrated Natural PG method. PMD has also close connections to state-of-the-art methods such as TRPO and PPO [50] which have achieved widespread empirical success [41, 42], including most recently in fine-tuning Large Language Models via RL from human feedback [35]. Interestingly, PMD has also been inspired by one of the most fundamental algorithms in RL: Policy Iteration (PI). While PI alternates between policy evaluation and policy improvement, PMD regularizes the latter step to address the instability issue of PI with inexact policy evaluation.

Policy Iteration and its variants have been extensively studied in the literature, see e.g. [5, 32, 39]. In particular, PI has been studied in conjunction with the lookahead mechanism [12], i.e. using multi-step greedy policy improvement instead of single-step greedy policies. Intuitively, the idea is that the application of the Bellman operator multiple times before computing a greedy policy leads to a more accurate approximation of the optimal value function. Implemented via Monte Carlo Tree Search (MCTS), multi-step greedy policy improvement[2] contributed to the empirical success of some of the

---

[*]Department of Computer Science, ETH Zürich, Switzerland. Contact: kprotopapas@student.ethz.ch, barakat9anas@gmail.com. Most of this work was completed when both authors were affiliated with ETH Zürich, K.P as a Master student and A.B. as a postdoctoral fellow. A.B. is currently affiliated with Singapore University of Technology and Design as a research fellow.

[2]Note that this is different from $n$-step return approaches which are rather used for policy evaluation [47].

most impressive applications of RL including AlphaGo, AlphaZero and MuZero [44, 46, 45, 40]. Besides this practical success, a body of work [12, 13, 49, 52, 51] has investigated the role of lookahead in improving the performance of RL algorithms, reporting a convergence rate speed-up when enhancing PI with $h$-step greedy policy improvement with a reasonable computational overhead.

In this work, we propose to cross-fertilize the PMD class of algorithms with multi-step greedy policy improvement to obtain a novel class of PMD algorithms enjoying the benefits of lookahead. Our contributions are as follows:

- We propose a novel class of algorithms called $h$-PMD enhancing PMD with multi-step greedy policy updates where $h$ is the depth of the lookahead. This class collapses to standard PMD when $h = 1$. When the stepsize parameter goes to infinity, we recover the PI algorithm with multiple-step greedy policy improvement previously analyzed in [12]. When solving a Markov Decision Process problem with $h$-PMD, we show that the value suboptimality gap converges to zero geometrically with a contraction factor of $\gamma^h$. This rate is faster than the standard convergence rate of PI and the similar rate for PMD which was recently established in [20] for $h \geq 2$. This rate improvement requires the computation of a $h$-step lookahead value function at each iteration which can be performed using planning methods such as tree-search. These results for exact $h$-PMD are exposed in Sec. 4, when value functions can be evaluated exactly.

- We examine the inexact setting where the $h$-step action value functions are not available. In this setting, we propose a Monte Carlo sample-based procedure to estimate the unknown $h$-step lookahead action-value function at each state-action pair. We provide a sample complexity for this Monte Carlo procedure, improving over standard PMD thanks to the use of lookahead. Larger lookahead depth translates into a better sample complexity and a faster suboptimality gap convergence rate. However, this improvement comes at a more intensive computational effort to perform the lookahead steps. Sec. 5 discusses this inexact setting and the aforementioned tradeoff.

- We extend our results to the function approximation setting to address the case of large state spaces where tabular methods are not tractable. In particular, we design a $h$-PMD algorithm where the $h$-step lookahead action-value function is approximated using a linear combination of state-action feature vectors in the policy evaluation step. Under linear function approximation and using a generative sampling model, we provide a performance bound for our $h$-PMD algorithm with linear function approximation. Our resulting sample complexity only depends on the dimension of the feature map space instead of the size of the state space and improves over prior work analyzing PMD with function approximation.

- We perform simulations to verify our theoretical findings empirically on the standard DeepSea RL environment from Deep Mind's *bsuite* [34]. Our experiments illustrate the convergence rate improvement of $h$-PMD with increasing lookahead depth $h$ in both the exact and inexact settings.

## 2 Preliminaries

**Notation.** For any integer $n \in \mathbb{N} \setminus \{0\}$, the set of integers from 1 to $n$ is denoted by $[n]$. For a finite set $\mathcal{X}$, we denote by $\Delta(\mathcal{X})$ the probability simplex over the set $\mathcal{X}$. For any $d \in \mathbb{N}$, we endow the Euclidean space $\mathbb{R}^d$ with the norm $\|\cdot\|_\infty$ defined for every $v \in \mathbb{R}^d$ by $\|v\|_\infty := \max_{i \in [d]} |v_i|$ where $v_i$ is the $i$-th coordinate. The relative interior of a set $\mathcal{Z}$ is denoted by $\mathrm{ri}(\mathcal{Z})$.

**Markov Decision Process (MDP).** We consider an infinite horizon discounted Markov Decision Process $\mathcal{M} = (\mathcal{S}, \mathcal{A}, r, \gamma, P, \rho)$ where $\mathcal{S}$ and $\mathcal{A}$ are finite sets of states and actions respectively with cardinalities $S = |\mathcal{S}|$ and $A = |\mathcal{A}|$ respectively, $P : \mathcal{S} \times \mathcal{A} \to \Delta(\mathcal{S})$ is the Markovian probability transition kernel, $r : \mathcal{S} \times \mathcal{A} \to [0, 1]$ is the reward function, $\gamma \in (0, 1)$ is the discount factor and $\rho \in \Delta(\mathcal{S})$ is the initial state distribution. A randomized stationary policy is a mapping $\pi : \mathcal{S} \to \Delta(\mathcal{A})$ specifying the probability $\pi(a|s)$ of selecting action $a$ in state $s$ for any $s \in \mathcal{S}, a \in \mathcal{A}$. The set of all such policies is denoted by $\Pi$. At each time step, the learning agent takes an action $a$ with probability $\pi(a|s)$, the environment transitions from the current state $s \in \mathcal{S}$ to the next state $s' \in \mathcal{S}$ with probability $P(s'|s, a)$ and the agent receives a reward $r(s, a)$.

**Value functions and optimal policy.** For any policy $\pi \in \Pi$, we define the state value function $V^\pi : \mathcal{S} \to \mathbb{R}$ for every $s \in \mathcal{S}$ by $V^\pi(s) = \mathbb{E}_\pi \left[ \sum_{t=0}^\infty \gamma^t r(s_t, a_t) | s_0 = s \right]$. The state-action value function $Q^\pi : \mathcal{S} \times \mathcal{A} \to \mathbb{R}$ can be similarly defined for every $s \in \mathcal{S}, a \in \mathcal{A}$ by $Q^\pi(s, a) = \mathbb{E}_\pi \left[ \sum_{t=0}^\infty \gamma^t r(s_t, a_t) | s_0 = s, a_0 = a \right]$ where $\mathbb{E}_\pi$ is the expectation over the state-

action Markov chain $(s_t, a_t)$ induced by the MDP and the policy $\pi$ generating actions. The goal is to find a policy $\pi$ maximizing $V^\pi$. A classic result shows that there exists an optimal deterministic policy $\pi^\star \in \Pi$ maximizing $V^\pi$ simultaneously for all the states [36]. In this work, we focus on searching for an $\epsilon$-optimal policy, i.e. a policy $\pi$ satisfying $\|V^\star - V^\pi\|_\infty \leq \epsilon$ where $V^\star := V^{\pi^\star}$.

**Bellman operators and Bellman equations.** We will often represent the reward function $r$ by a vector in $\mathbb{R}^{SA}$ and the transition kernel by the operator $P \in \mathbb{R}^{SA \times S}$ acting on vectors $v \in \mathbb{R}^S$ (which can also be seen as functions defined over $\mathcal{S}$) as follows: $(Pv)(s, a) = \sum_{s' \in \mathcal{S}} P(s'|s, a)v(s')$ for any $s \in \mathcal{S}, a \in \mathcal{A}$. We further define the mean operator $M^\pi : \mathbb{R}^{SA} \to \mathbb{R}^S$ mapping any vector $Q \in \mathbb{R}^{SA}$ to a vector in $\mathbb{R}^S$ whose components are given by $(M^\pi Q)(s) = \sum_{a \in \mathcal{A}} \pi(a|s)Q(s, a)$, and the maximum operator $M^\star : \mathbb{R}^{SA} \to \mathbb{R}^S$ defined by $(M^\star Q)(s) = \max_{a \in \mathcal{A}} Q(s, a)$ for any $s \in \mathcal{S}, Q \in \mathbb{R}^{SA}$. Using these notations, we introduce for any given policy $\pi \in \Pi$ the associated expected Bellman operator $\mathcal{T}^\pi : \mathbb{R}^S \to \mathbb{R}^S$ defined for every $V \in \mathbb{R}^S$ by $\mathcal{T}^\pi V := M^\pi(r + \gamma PV)$ and the Bellman optimality operator $\mathcal{T} : \mathbb{R}^S \to \mathbb{R}^S$ defined by $\mathcal{T}V := M^\star(r + \gamma PV)$ for every $V \in \mathbb{R}^S$. Using the notations $r^\pi := M^\pi r$ and $P^\pi := M^\pi P$, we can also simply write $\mathcal{T}^\pi V = r^\pi + \gamma P^\pi V$ and $\mathcal{T}V = \max_{\pi \in \Pi} \mathcal{T}^\pi V$ for any $V \in \mathbb{R}^S$.

The Bellman optimality operator $\mathcal{T}$ and the expected Bellman operator $\mathcal{T}^\pi$ for any given policy $\pi \in \Pi$ are $\gamma$-contraction mappings w.r.t. the $\|\cdot\|_\infty$ norm and hence admit unique fixed points $V^\star$ and $V^\pi$ respectively. In particular, these operators satisfy the following inequalities: $\|\mathcal{T}^\pi V - V^\pi\|_\infty \leq \gamma\|V - V^\pi\|_\infty$ and $\|\mathcal{T}V - V^\star\|_\infty \leq \gamma\|V - V^\star\|_\infty$ for any $V \in \mathbb{R}^S$. Moreover, we recall that for $\mathbf{e} = (1, 1, \ldots, 1) \in \mathbb{R}^S$ and any $c \in \mathbb{R}_{\geq 0}, V \in \mathbb{R}^S$, we have $\mathcal{T}^\pi(V + c\,\mathbf{e}) = \mathcal{T}^\pi V + \gamma c\,\mathbf{e}$ and $\mathcal{T}(V + c\,\mathbf{e}) = \mathcal{T}V + \gamma c\,\mathbf{e}$. We recall that the set $\mathcal{G}(V^\star) := \{\pi \in \Pi : \mathcal{T}^\pi V^\star = \mathcal{T}V^\star\}$ of 1-step greedy policies w.r.t. the optimal value $V^\star$ coincides with the set of stationary optimal policies.

# 3 A Refresher on PMD and PI with Lookahead

## 3.1 PMD and its Connection to PI

Policy Gradient methods consist in performing gradient ascent updates with respect to an objective $\mathbb{E}_\rho[V^\pi(s)]$ w.r.t. the policy $\pi$ parametrized by its values $\pi(a|s)$ for $s, a \in \mathcal{S} \times \mathcal{A}$ in the case of a tabular policy parametrization. Interestingly, policy gradient methods have been recently shown to be closely connected to a class of Policy Mirror Descent algorithms by using dynamical reweighting of the Bregman divergence with discounted visitation distribution coefficients in a classical mirror descent update rule using policy gradients. We refer the reader to section 4 in [53] or section 3.1 in [20] for a detailed exposition. The resulting PMD update rule is given by

$$\pi_s^{k+1} \in \text{argmax}_{\pi_s \in \Delta(\mathcal{A})} \left\{ \eta_k \langle Q_s^{\pi_k}, \pi_s \rangle - D_\phi(\pi_s, \pi_s^k) \right\}, \tag{PMD}$$

where we use the shorthand notations $\pi_s^k = \pi_k(\cdot|s)$ and $\pi_s = \pi(\cdot|s)$, where $(\eta_k)$ is a sequence of positive stepsizes, $Q_s^{\pi_k} \in \mathbb{R}^A$ is a vector containing state action values for $\pi_k$ at state $s$, and $D_\phi$ is the Bregman divergence induced by a mirror map $\phi : \text{dom}\,\phi \to \mathbb{R}$ such that $\Delta(\mathcal{A}) \subset \text{dom}\,\phi$, i.e. for any $\pi, \pi' \in \Pi, s \in \mathcal{S}$,

$$D_\phi(\pi_s, \pi_s') = \phi(\pi_s) - \phi(\pi_s') - \langle \nabla\phi(\pi_s'), \pi_s - \pi_s' \rangle,$$

where we suppose throughout this paper that the function $\phi$ is of Legendre type, i.e. strictly convex and essentially smooth in the relative interior of $\text{dom}\,\phi$ (see [37], section 26). The mirror map choice gives rise to a large class of algorithms. Two special cases are noteworthy: (a) When $\phi$ is the squared 2-norm, the Bregman divergence is the squared Euclidean distance and the resulting algorithm is a projected $Q$-ascent and (b) when $\phi$ is the negative entropy, the corresponding Bregman divergence is the Kullback-Leibler (KL) divergence and (PMD) becomes the popular Natural Policy Gradient algorithm (see e.g. [53], section 4 for details).

Now, using our operator notations, one can observe that the (PMD) update rule above is equivalent to the following update rule (see Lemma A.1 for a proof),

$$\pi_{k+1} \in \text{argmax}_{\pi \in \Pi} \left\{ \eta_k \mathcal{T}^\pi V^{\pi_k} - D_\phi(\pi, \pi_k) \right\}, \tag{1}$$

where $D_\phi(\pi, \pi_k) \in \mathbb{R}^S$ is a vector whose components are given by $D_\phi(\pi_s, \pi_s^k)$ for $s \in \mathcal{S}$. Note that the maximization can be carried out independently for each state component.

**Connection to Policy Iteration.** As previously noticed in the literature [53, 20], PMD can be interpreted as a 'soft' PI. Indeed, taking infinite stepsizes $\eta_k$ in PMD (or a null Bregman divergence) immediately leads to the following update rule:

$$\pi_{k+1} \in \operatorname{argmax}_{\pi \in \Pi} \left\{ \mathcal{T}^\pi V^{\pi_k} \right\} , \tag{PI}$$

which corresponds to synchronous Policy Iteration (PI). The method alternates between a policy evaluation step to estimate the value function $V^{\pi_k}$ of the current policy $\pi_k$ and a policy improvement step implemented as a one-step greedy policy with respect to the current value function estimate.

### 3.2 Policy Iteration with Lookahead

As discussed in [12], Policy Iteration can be generalized by performing $h \geq 1$ greedy policy improvement steps at each iteration instead of a single step. The resulting $h$-PI update rule is:

$$\pi_{k+1} \in \operatorname{argmax}_{\pi \in \Pi} \left\{ \mathcal{T}^\pi \mathcal{T}^{h-1} V^{\pi_k} \right\} , \tag{$h$-PI}$$

where $h \in \mathbb{N} \setminus \{0\}$. The $h$-greedy policy $\pi^{k+1}$ w.r.t $V^{\pi_k}$ selects the first optimal action of a non-stationary $h$-horizon optimal control problem. It can also be seen as the 1-step greedy policy w.r.t. $\mathcal{T}^{h-1} V^{\pi_k}$. In the rest of this paper, we will denote by $\mathcal{G}_h(V)$ the set of $h$-step greedy policies w.r.t. any value function $V \in \mathbb{R}^S$, i.e. $\mathcal{G}_h(V) := \operatorname{argmax}_{\pi \in \Pi} \mathcal{T}^\pi \mathcal{T}^{h-1} V = \{\pi \in \Pi : \mathcal{T}^\pi \mathcal{T}^{h-1} V = \mathcal{T}^h V\}$. Notice that $\mathcal{G}_1(V) = \mathcal{G}(V)$. Overall, the $h$-PI method alternates between finding a $h$-greedy policy and estimating the value of this policy. Interestingly, similarly to PI, a $h$-greedy policy guarantees monotonic improvement, i.e. $V^{\pi'} \geq V^\pi$ component-wise for any $\pi \in \Pi, \pi' \in \mathcal{G}_h(V^\pi)$. Moreover, since $\mathcal{T}^h$ is a $\gamma^h$-contraction, it can be shown that the sequence $(\|V^\star - V^{\pi_k}\|_\infty)_k$ is contracting with coefficient $\gamma^h$. See section 3 in [12] for further details about $h$-PI.

*Remark* 3.1. While the iteration complexity always improves with larger depth $h$ (see Theorem 3 in [12]), each iteration becomes more computationally demanding. As mentioned in [14], when the model is known, the $h$-greedy policy can be computed with Dynamic Programming (DP) in linear time in the depth $h$.

Another possibility is to implement a tree-search of depth $h$ starting from the root state $s$ to compute the $h$-greedy policy in deterministic MDPs. We refer to [14] for further details. Sampling-based tree search methods such as Monte Carlo Tree Search (MCTS) (see [9]) can be used to circumvent the exponential complexity in the depth $h$ of tree search methods.

## 4 Policy Mirror Descent with Lookahead

In this section we propose our novel Policy Mirror Descent (PMD) algorithm which incorporates $h$-step lookahead for policy improvement: $h$-PMD.

### 4.1 $h$-PMD: Using $h$-Step Greedy Updates

Similarly to the generalization of PI to $h$-PI discussed in the previous section, we obtain our algorithm $h$-PMD by incorporating lookahead to PMD as follows:

$$\pi_{k+1} = \operatorname{argmax}_{\pi \in \Pi} \left\{ \eta_k \mathcal{T}^\pi \mathcal{T}^{h-1} V^{\pi_k} - D_\phi(\pi, \pi_k) \right\} . \tag{$h$-PMD}$$

Notice that we recover (1), i.e. (PMD), by setting $h = 1$. We now provide an alternative way to write the $h$-PMD update rule which is more convenient for its implementation. We introduce a few additional notations for this purpose. For any policy $\pi \in \Pi$, let $V_h^\pi := \mathcal{T}^{h-1} V^\pi$ for any integer $h \geq 1$. Consider the corresponding action value function defined for every $s, a \in \mathcal{S} \times \mathcal{A}$ by:

$$Q_h^\pi(s, a) := r(s, a) + \gamma (P V_h^\pi)(s, a) . \tag{2}$$

Using these notations, it can be easily shown that the $h$-PMD rule above is equivalent to:

$$\pi_s^{k+1} = \operatorname{argmax}_{\pi \in \Pi} \left\{ \eta_k \langle Q_h^{\pi_k}(s, \cdot), \pi_s \rangle - D_\phi(\pi_s, \pi_s^k) \right\} . \tag{$h$-PMD'}$$

Before moving to the analysis of this scheme, we provide two concrete examples for the implementation of $h$-PMD' corresponding to the choice of two standard mirror maps.

(1) *Projected $Q_h$-ascent.* When the mirror map $\phi$ is the squared 2-norm, the corresponding Bregman divergence is the squared Euclidean distance and $h$-PMD' becomes

$$\pi_s^{k+1} = \text{Proj}_{\Delta(\mathcal{A})}(\pi_s^k + \eta_k Q_h^{\pi_k}(s, \cdot)), \tag{3}$$

for every $s \in \mathcal{S}$ and $\text{Proj}_{\Delta(\mathcal{A})}$ is the projection operator on the simplex $\Delta(\mathcal{A})$.

(2) *Multiplicative $Q_h$-ascent.* When the mirror map $\phi$ is the negative entropy, the Bregman divergence $D_\phi$ is the Kullback-Leibler divergence and for every $(s, a) \in \mathcal{S} \times \mathcal{A}$,

$$\pi_{k+1}(a|s) = \pi_k(a|s)\frac{\exp(\eta_k Q_h^{\pi_k}(s, a))}{Z_k(s)}, \quad Z_k(s) := \sum_{a \in \mathcal{A}} \pi_k(a|s)\exp(\eta_k Q_h^{\pi_k}(s, a)). \tag{4}$$

**Connection to AlphaZero.** Before switching gears to the analysis, we point out an interesting connection between $h$-PMD and the successful class of AlphaZero algorithms [46, 45, 40] which are based on heuristics.

It has been shown in [16] that AlphaZero can be seen as a regularized policy optimization algorithm, drawing a connection to the standard PMD algorithm (with $h = 1$). We argue that AlphaZero is even more naturally connected to our $h$-PMD algorithm. Indeed, the $Q$ values in [16, Eq. (7) p. 3] correspond more closely to the lookahead action value function $Q_h^\pi$ approximated via tree search instead of the standard $Q^\pi$ function with $h = 1$. This connection clearly delineates the dependence on the lookahead depth (or tree search depth) $h$ which was not clear in [16]. We have implemented a version of our $h$-PMD algorithm with Deep Mind's MCTS implementation (see section D.6 and the code provided for details). Conducting further experiments to compare our algorithm to AlphaZero on similar large scale settings would be interesting. We are not aware of any theoretical convergence guarantee for AlphaZero which relies on many heuristics. In contrast, our $h$-PMD algorithm enjoys theoretical guarantees that we establish in the next sections.

## 4.2 Convergence Analysis for Exact $h$-PMD

Using the contraction properties of the Bellman operators, it can be shown that the suboptimality gap $\|V^{\pi_k} - V^*\|_\infty$ for PI iterates converges to zero at a linear rate with a contraction factor $\gamma$, regardless of the underlying MDP. This instance-independent convergence rate was generalized to PMD in [20]. In this section, we establish a linear convergence rate for the subobtimality value function gap of the exact $h$-PMD algorithm with a contraction factor of $\gamma^h$ where $h$ is the lookahead depth. We assume for now that the value function $V^{\pi_k}$ and a greedy policy $\pi_{k+1}$ in ($h$-PMD) can be computed exactly. These assumptions will be relaxed in the next section dealing with inexact $h$-PMD.

**Theorem 4.1** (Exact $h$-PMD). *Let $(c_k)$ be a sequence of positive reals and let the stepsize $\eta_k$ in ($h$-PMD) satisfy $\eta_k \geq \frac{1}{c_k}\|\min_{\pi \in \mathcal{G}_h(V^{\pi_k})} D_\phi(\pi, \pi_k)\|_\infty$ where we recall that $\mathcal{G}_h(V^{\pi_k})$ is the set of greedy policies with respect to $\mathcal{T}^{h-1}V^{\pi_k}$ and that the minimum is computed component-wise. Initialized at $\pi_0 \in \text{ri}(\Pi)$, the iterates $(\pi_k)$ of ($h$-PMD) with $h \in \mathbb{N} \setminus \{0\}$ satisfy for every $k \in \mathbb{N}$,*

$$\|V^\star - V^{\pi_k}\|_\infty \leq \gamma^{hk}\left(\|V^\star - V^{\pi_0}\|_\infty + \frac{1}{1 - \gamma}\sum_{t=1}^{k}\frac{c_{t-1}}{\gamma^{ht}}\right).$$

A few remarks are in order regarding this result:

- Compared to [20], our new algorithm achieves a faster $\gamma^h$-rate where $h$ is the depth of the lookahead. It should be noted that the approach in [20] is claimed to be optimal over all methods that are guaranteed to increase the probability of the greedy action at each timestep. Adding lookahead to the policy improvement step circumvents this restriction.

- Unlike prior work [53, 25] featuring distribution mismatch coefficients which can scale with the size of the state space, our convergence rate does not depend on any other instance-dependent quantities. This is thanks to the analysis which circumvents the use of the performance difference lemma similarly to [20].

- Choosing $c_t = \gamma^{2h(t+1)}c_0$ for a positive constant $c_0$ for every integer $t$, the bound becomes (after bounding the geometric sum): $\|V^\star - V^{\pi_k}\|_\infty \leq \gamma^{hk}\left(\|V^\star - V^{\pi_0}\|_\infty + \frac{c_0}{(1-\gamma)^2}\right)$. As $c_0 \to 0$ we recover the linear convergence result of $h$-PI established in [12].

- This faster rate comes at the cost of a more complex value function computation at each iteration. However, our experiments suggest that the benefits of the faster convergence rate greatly outweigh the extra cost of computing the lookahead, in terms of both overall running time until convergence and sample complexity (in the inexact case). See section 7 and Appendix D for evidence.

- The cost of computing the adaptive stepsizes is typically simply that of computing a Bregman divergence between two policies. See Appendix A.3, D.4 for a more detailed discussion.

- We defer the proof of Theorem 4.1 to Appendix A. Our proof highlights the relationship between $h$-PMD and $h$-PI through the explicit use of Bellman operators. Notice that even in the particular case of $h = 1$, our proof is more compact compared to the one in [20] (see sec. 6 and Lemma A.1 to A.3 therein for comparison) using our convenient notations.

## 5 Inexact and Stochastic $h$-Policy Mirror Descent

In this section, we discuss the case where the lookahead action value $Q_h^{\pi_k}$ defined in (2) is unknown. We propose a procedure to estimate it using Monte Carlo sampling and we briefly discuss an alternative using a tree search method. The resulting inexact $h$-PMD update rule for every $s \in \mathcal{S}$ is

$$\pi_s^{k+1} = \operatorname{argmax}_{\pi_s \in \Delta(\mathcal{A})} \left\{ \eta_k \langle \hat{Q}_h^{\pi_k}(s, \cdot), \pi_s \rangle - D_\phi(\pi_s, \pi_s^k) \right\}, \tag{5}$$

where $\hat{Q}_h^{\pi_k}$ is the estimated version of the lookahead action value $Q_h^{\pi_k}$ induced by policy $\pi_k$. We conclude this section by providing a convergence analysis for inexact $h$-PMD and discussing its sample complexity using the proposed Monte Carlo estimator.

### 5.1 Monte Carlo $h$-Greedy Policy Evaluation

Estimating the lookahead action value function $Q_h^\pi$ for a given policy $\pi$ involves solving a $h$-horizon planning problem using samples from the MDP. Our estimation procedure combines a standard planning method with Monte Carlo estimation under a generative model of the MDP. We give an algorithmic description of the procedure below, and defer the reader to Appendix B.1 for a precise definition of the procedure. Applying the recursive algorithm below with $k := h$ returns an estimate for the action value function $Q_h^\pi(s, a)$ at a given state-action pair $(s, a) \in \mathcal{S} \times \mathcal{A}$.

---

**Algorithm 1** Lookahead Q-function Estimation via Monte Carlo Planning

> **Procedure** $Q(k, s, a, \pi)$
> **if** $k = 1$ **then**
>     **return** $r(s, a) + \frac{\gamma}{M} \sum_{i=1}^M \hat{V}^\pi(s_i')$ (where $s_i' \sim \mathcal{P}(\cdot|s, a)$ for $i \in [M]$)
> **else**
>     **return** $r(s, a) + \frac{\gamma}{M} \sum_{i=1}^M \max_{a' \in \mathcal{A}} Q(k - 1, s_i', a', \pi)$ (where $s_i' \sim \mathcal{P}(\cdot|s, a)$ for $i \in [M]$)
> **end if**

---

Note that the base case of the recursion estimates the value function using Monte Carlo rollouts, see Appendix B.1.

*Remark* 5.1. Notice that actions are exhaustively selected in our procedure like in a planning method. Using bandit ideas to guide Monte Carlo planning (see e.g. [23]) would be interesting to investigate for more efficiency. We leave the investigation of such selective action sampling and exploration for future work. In practice, lookahead policies are often computed using tree search techniques [12].

*Remark* 5.2. When $h = 1$, the procedure collapses to a simple Monte Carlo estimate of the Q-value function $Q^{\pi_k}$ at each iteration $k$ of the $h$-PMD algorithm.

### 5.2 Analysis of Inexact $h$-PMD

The next theorem is a generalization of Theorem 4.1 to the inexact setting in which the lookahead function $Q_h^{\pi_k}$ can be estimated with some errors at each iteration $k$ of $h$-PMD.

**Theorem 5.3** (Inexact $h$-PMD). *Suppose there exists $b \in \mathbb{R}_+$ s.t. $\|\hat{Q}_h^{\pi_k} - Q_h^{\pi_k}\|_\infty \leq b$ where the maximum norm is over both the state and action spaces. Let $\tilde{\mathcal{G}}_h = \operatorname{argmax}_{\pi \in \Pi} M^\pi \hat{Q}_h^{\pi_k}$ and let $(c_k)$ be a sequence of positive reals and let the stepsize $\eta_k$ in ($h$-PMD) satisfy $\eta_k \geq \frac{1}{c_k} \|\min_{\pi \in \tilde{\mathcal{G}}_h} D_\phi(\pi, \pi_k)\|_\infty$*

*where we recall that the minimum is computed component-wise. Initialized at $\pi_0 \in \mathrm{ri}(\Pi)$, the iterates $(\pi_k)$ of inexact (h-PMD) with $h \in \mathbb{N} \setminus \{0\}$ satisfy for every $k \in \mathbb{N}$,*

$$\|V^\star - V^{\pi_k}\|_\infty \leq \gamma^{hk} \left( \|V^\star - V^{\pi_0}\|_\infty + \frac{1}{1-\gamma} \sum_{t=1}^{k} \frac{c_{t-1}}{\gamma^{ht}} \right) + \frac{2b}{(1-\gamma)(1-\gamma^h)} \ .$$

The proof of this result is similar to the proof of Theorem 4.1 and consists in propagating the error $b$ in the analysis. We defer it to Appendix B.

As can be directly seen from this formulation, the $\gamma^h$ convergence rate from the exact setting generalizes to the inexact setting. Therefore, higher lookahead depths lead to faster convergence rates. Another improvement with relation to [20] is that the additive error term has a milder dependence on the effective horizon $1 - \gamma$: the term $\frac{2b}{(1-\gamma)(1-\gamma^h)}$ is smaller for all values of $h \geq 2$ than $\frac{4b}{(1-\gamma)^2}$. Larger lookahead depths yield a smaller asymptotic error in terms of the suboptimality gap.

We are now ready to establish the sample complexity of inexact $h$-PMD under a generative model using Theorem 5.3 together with concentration results for our Monte Carlo lookahead action value estimator.

**Theorem 5.4** (Sample complexity of $h$-PMD). *Assume that inexact $h$-PMD is run for a number of iterations $K > \frac{1}{h(1-\gamma)} \log(\frac{4}{\epsilon(1-\gamma)(1-\gamma^h)})$, using the choice of stepsize defined by the sequence $(c_k) := (\gamma^{2h(k+1)})$. Additionally, suppose we are given a target suboptimality value $\epsilon > 0$, and a probability threshold $\delta > 0$. Finally, assume the lookahead value function $\hat{Q}_h^{\pi_k}$ is approximated at each iteration with the Monte Carlo estimation procedure described in section 5.1, with the following parameter values: $M_0 = \tilde{\mathcal{O}}(\frac{\gamma^{2h}}{(1-\gamma)^4(1-\gamma^h)^2\varepsilon^2})$ and for all $j \in [h]$, $M_j = M = \tilde{\mathcal{O}}(\frac{1}{(1-\gamma)^6\varepsilon^2})$. Then, with probability at least $1 - \delta$, the suboptimality at all iterations $k$ satisfy the following bound:*

$$\|V^\star - V^{\pi_k}\|_\infty \leq \gamma^{hk} \left( \|V^\star - V^{\pi_0}\|_\infty + \frac{1-\gamma^{hk}}{(1-\gamma)(1-\gamma^h)} \right) + \epsilon \tag{6}$$

*Using this procedure, $h$-PMD uses at most $KM_0HS + hKMSA$ samples in total. The overall sample complexity of the inexact $h$-PMD is then given by $\tilde{\mathcal{O}}(\frac{S}{h\epsilon^2(1-\gamma)^6(1-\gamma^h)^2} + \frac{SA}{\epsilon^2(1-\gamma)^7})$ where the notation $\tilde{\mathcal{O}}$ hides at most polylogarithmic factors.*

Compared to Theorem 5.1 in [20], the dependence on the effective horizon improves by a factor of the order of $1/(1-\gamma)$ for $h > A^{-1}(1-\gamma)^{-1}$ thanks to our Monte Carlo lookahead estimator. Their sample complexity is of order $\tilde{\mathcal{O}}(\frac{1}{\epsilon^2(1-\gamma)^8})$, whereas for $h > \frac{1}{A(1-\gamma)}$ ours becomes $\tilde{\mathcal{O}}(\frac{1}{\epsilon^2(1-\gamma)^7})$.

Our sample complexity does not depend on quantities such as the distribution mismatch coefficient used in prior work [53, 25], which may scale with the state space size.

# 6  Extension to Function Approximation

In MDPs with prohibitively large state action spaces, we represent each state-action pair $(s, a) \in \mathcal{S} \times \mathcal{A}$ by a feature vector $\psi(s, a) \in \mathbb{R}^d$ where typically $d \ll SA$ and where $\psi : \mathcal{S} \times \mathcal{A} \to \mathbb{R}^d$ is a feature map, also represented as a matrix $\Psi \in \mathbb{R}^{SA \times d}$. Assuming that this feature map holds enough "information" about each state action pair (more rigorous assumptions will be provided later), it is possible to approximate any value function using a matrix vector product $\Psi\theta$, where $\theta \in \mathbb{R}^d$. In this section, we propose an inexact $h$-PMD algorithm using action value function approximation.

## 6.1  Inexact $h$-PMD with Function Approximation

Using the notations above, approximating $Q_h^{\pi_k}$ by $\Psi\theta_k$, the inexact $h$-PMD update rule becomes:

$$\pi_s^{k+1} \in \mathrm{argmax}_{\pi_s \in \Delta(\mathcal{S})} \left\{ \eta_k \langle (\Psi\theta_k)_s, \pi_s \rangle - D_\phi(\pi_s, \pi_s^k) \right\} \ . \tag{7}$$

We note that the policy update above can be implicit to avoid computing a policy for every state. It is possible to implement this algorithm using our Monte Carlo planning method in a similar fashion

to [27]: use the procedure in section 5.1 to estimate $Q_h^{\pi_k}(s, a)$ for all state action pairs $(s, a)$ in some set $\mathcal{C} \subset \mathcal{S} \times \mathcal{A}$, and use these estimates as targets for a least squares regression to approximate $Q_h^{\pi_k}$ by $\Psi \theta_k$ at time step $k$. See Appendix C for further details. We conclude this section with a convergence analysis of $h$-PMD with linear function approximation.

## 6.2  Convergence Analysis of $h$-PMD with Linear Function Approximation

Our analysis follows the approach of [27]. We make the following standard assumptions.

**Assumption 6.1.** The feature matrix $\Psi \in \mathbb{R}^{SA \times d}$ where $d \leq SA$ is full rank.

**Assumption 6.2** (Approximate Universal value function realizability)**.** There exists $\epsilon_{\text{FA}} > 0$ s.t. for any $\pi \in \Pi$, $\inf_{\theta \in \mathbb{R}^d} \|Q_h^\pi - \Psi\theta\|_\infty \leq \epsilon_{\text{FA}}$.

We defer a discussion about Assumption 6.2 to Appendix C.2.

**Theorem 6.3** (Convergence of $h$-PMD with linear function approximation)**.** *Let $(\theta_k)$ be the sequence of iterates produced by $h$-PMD with linear function approximation, run for $K$ iterations, starting with policy $\pi_0 \in \text{ri}(\Pi)$, using stepsizes defined by $(c_k)$. Assume that targets $\hat{Q}_h^{\pi_k}$ were computed using the procedure described in sec. 5.1 such that with probability at least $1 - \delta$, $\forall k \leq K$, $\forall z \in \mathcal{C} \subset \mathcal{S} \times \mathcal{A}$ the targets satisfy $|\hat{Q}_h^{\pi_k}(z) - Q_h^{\pi_k}(z)| \leq \epsilon$.*[3]

*Then, there exists a choice of $\mathcal{C}$ for which the policy iterates satisfy at each iteration:*

$$\|V^\star - V^{\pi_k}\|_\infty \leq \gamma^{hk} \left( \|V^\star - V^{\pi_0}\|_\infty + \frac{1}{1-\gamma} \sum_{t=1}^k \frac{c_{t-1}}{1-\gamma} \right) + \frac{2\sqrt{d}\,\epsilon + 2\left(1 + \sqrt{d}\right)\epsilon_{FA}}{(1-\gamma)(1-\gamma^h)}.$$

The proof of Theorem 6.3 can be found in Appendix C.3. Notice that our performance bound does not depend on the state space size like in sec. 5 and depends on the dimension $d$ of the feature space instead. We should notice though that the choice of the set $\mathcal{C}$ is crucial for this result and given by the Kiefer-Wolfowitz theorem [22] which guarantees that $|\mathcal{C}| = \mathcal{O}(d^2)$ as discussed in [27]. Moreover, our method for function approximation yields a sample complexity that is instance independent. Existing results for PMD with function approximation [3, 54] depend on distribution mismatch coefficients which can scale with the size of the state space.

In terms of computational complexity, at each iteration, $h$-PMD uses $O((MA)^h)$ elementary operations where $M$ is the size of the minibatch of trajectories used, $A$ is the size of the action space and $h$ is the lookahead depth. This computational complexity which scales exponentially in $h$ is inherent to tree search methods. Despite this seemingly prohibitive computational scaling, tree search has been instrumental in practice for achieving state of the art results in some environments [40, 19] and modern implementations (notably using GPU-based search on vectorized environments) make tree search much more efficient in practice.

## 7  Simulations

We conduct simulations to investigate the effect of the lookahead depth on the convergence rate and illustrate our theoretical findings. We run the $h$-PMD algorithm for different values of $h$ in both exact and inexact settings on the DeepSea environment from DeepMind's *bsuite* [34] using a grid size of 64 by 64, and a discount factor $\gamma = 0.99$. Additional experiments are provided in Appendix D. Our codebase where all our experiments can be replicated is available here: `https://gitlab.com/kimon.protopapa/pmd-lookahead`.

**Exact $h$-PMD.** We run the exact $h$-PMD algorithm for 100 iterations for increasing values of $h$ using the KL divergence. Similar results were observed for the Euclidean divergence. We tested two different stepsize schedules: (a) in dotted lines in Fig. 1 (left), $\eta_k$ equal to its lower bound in sec. 4, with the choice $c_k := \gamma^{2h(k+1)}$ (note the dependence on $h$); and (b) in solid lines, $\eta_k$ identical stepsize schedule across all values of $h$ with $c_k := \gamma^{2(k+1)}$ to isolate the effect of the lookahead. We clearly observe under both stepsize schedules that $h$-PMD converges in fewer iterations when lookahead value functions are used instead of regular $Q$ functions.

---

[3]This can be achieved by choosing the parameters as in Theorem 5.4 (see Appendix C for details).

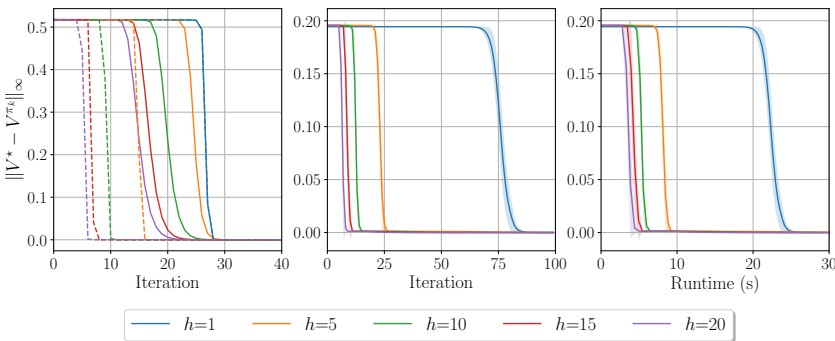

Figure 1: Suboptimality value function gap for $h$-PMD in the exact (left) and inexact (middle/right) settings, plotted against iterations in the exact case (left) and against both iterations (middle) and runtime (right) in the inexact case. 16 runs performed for each $h$, mean in solid line and standard deviation as shaded area. In dotted lines (left), the step size $\eta_k$ is equal to its lower bound in sec. 4, with the choice $c_k := \gamma^{2h(k+1)}$ (note the dependence on $h$) and in solid lines, the step size $\eta_k$ is set using an identical stepsize schedule across all values of $h$ with $c_k := \gamma^{2(k+1)}$ to isolate the effect of the lookahead. Notice that higher values of $h$ still perform better even in terms of runtime.

**Inexact $h$-PMD.** In this setting, we estimated the value function $Q_h$ using the vanilla Monte Carlo planning procedure detailed in section 5 in a stochastic variant of the DeepSea environment, and all the parameters except for $h$ were kept identical across runs. As predicted by our results, $h$-PMD converges in fewer iterations when $h$ is increased (see Fig. 1 (middle)). We also observed in our simulations that $h$-PMD uses less samples overall (see Appendix D for the total number of samples used at each iteration), and usually converges after less overall computation time (see Fig. 1 (right)). We also refer the reader to Appendix D where we have performed additional experiments with a larger lookahead depth $h = 100$. In this case, the algorithm converges in a single iteration. This is theoretically expected as computing the lookahead values with very large $h$ boils down to computing the optimal values like in value iteration with a large number of iterations. We also performed additional experiments in continuous control tasks to illustrate the general applicability of our algorithm (see Fig. 8 in Appendix D).

*Remark* 7.1. (**Choice of the lookahead depth $h$.**) The lookahead depth is a hyperparameter of the algorithm and can be tuned similarly to other hyperparameters such as the step size of the algorithm. Of course, the value would depend on the environment and the structure of the reward at hand. Sparse and delayed reward settings will likely benefit from lookahead with larger depth values. We have performed several simulations with different values of $h$ for each environment setting and the performance can potentially improve drastically with a better lookahead depth value (see also appendix D for further simulations). In addition, we observe that larger lookahead depth is not always better: see Fig. 7 in the appendix for an example where large lookahead depth becomes slower and does not perform better. Note that in this more challenging practical setting the best performance is not obtained for higher values of $h$: intermediate values of $h$ perform better. This illustrates the tradeoff in choosing the depth $h$ between an improved convergence rate and the computational cost induced by a larger $h$. We believe further investigation regarding the selection of the depth parameter might be useful to further improve the practical performance of the algorithm.

## 8    Related Work

**Policy Mirror Descent and Policy Gradient Methods.** Motivated by their empirical success [41, 42], the analysis of PG methods has recently attracted a lot of attention [1, 6, 21, 7, 53, 25, 20]. Among these works, a line of research focused on the analysis of PMD [53, 25, 28, 29, 20, 3] as a flexible algorithmic framework, and its particular instances such as the celebrated natural policy gradient [21, 54]. In particular, for solving unregularized MDPs (which is our main focus), Xiao [53] established linear convergence of PMD in the tabular setting with a rate depending on the instance dependent distribution mismatch coefficients, and extended their results to the inexact setting using a generative model. Similar results were established for the particular case of the natural policy gradient algorithm [6, 21]. More recently, Johnson, Pike-Burke, and Rebeschini [20] showed a dimension-free

rate for PMD with exact policy evaluation using adaptive stepsizes, closing the gap with PI. Notably, they further show that this rate is optimal and their analysis inspired from PI circumvents the use of the performance difference lemma which was prevalent in prior work. We improve over these results using lookahead and we extend our analysis beyond the tabular setting by using linear function approximation. To the best of our knowledge, the use of multi-step greedy policy improvement in PMD and its analysis are novel in the literature.

**Policy Iteration with Multiple-Step Policy Improvement.** Multi-step greedy policies have been successfully used in applications such as the game of Go and have been studied in a body of works [12, 13, 52, 15, 49]. In particular, multi-step greedy policy improvement has been studied in conjunction with PI in [12, 52]. Efroni et al. [12] introduced $h$-PI which incorporates $h$-lookahead to PI and generalized existing analyses on PI with 1-step greedy policy improvement to $h$-step greedy policies. However, their work requires access to an $h$-greedy policy oracle, and the $h$-PI algorithm is unstable in the stochastic setting even for $h = 1$ because of potentially large policy updates. We address all these issues in the present work with our $h$-PMD algorithm. Winnicki and Srikant [52] showed that a first-visit version of a PI scheme using a single sample path for policy evaluation converges to the optimal policy provided that the policy improvement step uses lookahead instead of a 1-step greedy policy. Unlike our present work, the latter work assumes access to a lookahead greedy policy oracle. More recently, Alegre et al. [2] proposed to use lookahead for policy improvement in a transfer learning setting. Extending Generalized Policy Improvement (GPI) [4] which identifies a new policy that simultaneously improves over all previously learned policies each solving a specific task, they introduce $h$-GPI which is a multi-step extension of GPI. We rather consider a lookahead version of PMD including policy gradient methods instead of PI. We focus on solving a single task and provide policy optimization guarantees in terms of iteration and sample complexities.

**Lookahead-Based Policy Improvement with Tree Search.** Lookahead policy improvement is usually implemented via tree search methods [9, 43, 40, 24, 18, 14]. Recently, Dalal et al. [11] proposed a method combining policy gradient and tree search to reduce the variance of stochastic policy gradients. The method relies on a softmax policy parametrization incorporating a tree expansion. Our general $h$-PMD framework which incorporates a different lookahead policy improvement step also brings together PG and tree search methods. Morimura et al. [31] designed a method with a mixture policy of PG and MCTS for non-Markov Decision Processes. Grill et al. [17] showed that AlphaZero as well as several other MCTS algorithms compute approximate solutions to a family of regularized policy optimization problems which bear similarities with our $h$-PMD algorithm update rule for $h = 1$ (see sec. 3.2 therein).

## 9 Conclusion

In this work, we introduced a novel class of PMD algorithms with lookahead inspired by the success of lookahead policies in practice. We have shown an improved $\gamma^h$-linear convergence rate depending on the lookahead depth in the exact setting. We proposed a stochastic version of our algorithm enjoying an improved sample complexity. We further extended our results to scale to large state spaces via linear function approximation with a performance bound independent of the state space size. Our paper offers several interesting directions for future research. A possible avenue for future work is to investigate the use of more general function approximators such as neural networks to approximate the lookahead value functions and scale to large state-action spaces. Recent work for PMD along these lines [3] might be a good starting point. Enhancing our algorithm with exploration mechanisms (see e.g. the recent work [33]) is an important future direction. Our PMD update rule might offer some advantage in this regard as was recently observed in the literature [28]. Constructing more efficient fully online estimators for the lookahead action values using MCTS and designing adaptive lookahead strategies to select the tree search horizon [38] are promising avenues for future work. Our algorithm brings together two popular and successful families of RL algorithms: PG methods and tree search methods. Looking forward, we hope our work further stimulates research efforts to investigate practical and efficient methods enjoying the benefits of both classes of methods.

## Acknowledgments and Disclosure of Funding

We would like to thank the anonymous area chair and reviewers for their useful comments which helped us to improve our paper. Most of this work was completed when A.B. was affiliated with ETH Zürich as a postdoctoral fellow. This work was partially supported by an ETH Foundations of Data Science (ETH-FDS) postdoctoral fellowship.

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

# Contents

# A Additional Details and Proofs for Section 4: Exact $h$-PMD

**Lemma A.1.** *For any policies* $\pi, \pi' \in \Pi$ *and every state* $s \in \mathcal{S}$, *we have* $\langle Q_s^{\pi'}, \pi_s \rangle = \mathcal{T}^\pi V^{\pi'}(s)$. *As a consequence, the update rules (PMD) and* (1) *in the main part are equivalent.*

*Proof.* Let $\pi, \pi' \in \Pi$. For every $s \in \mathcal{S}$, we have

$$\langle Q_s^{\pi'}, \pi_s \rangle = \sum_{a \in \mathcal{A}} Q^{\pi'}(s, a)\, \pi(a|s) \qquad \text{(by definition of the scalar product)}$$

$$= \sum_{a \in \mathcal{A}} \left( r(s, a) + \gamma \sum_{s' \in \mathcal{S}} P(s'|s, a) V^{\pi'}(s') \right) \pi(a|s) \qquad (Q^{\pi'} \text{ as a function of } V^{\pi'})$$

$$= r^\pi(s) + \gamma \sum_{s' \in \mathcal{S}} \left( \sum_{a \in \mathcal{A}} P(s'|s, a)\pi(a|s) \right) V^{\pi'}(s')$$

$$\text{(by rearranging and using the definition of } r^\pi)$$

$$= r^\pi(s) + \gamma P^\pi V^{\pi'}(s) \qquad \text{(using the definition of } P^\pi)$$

$$= \mathcal{T}^\pi V^{\pi'}(s). \tag{8}$$

The second part of the statement holds by setting $\pi' = \pi_k$.

$\square$

## A.1 Auxiliary Lemma for the Proof of Theorem 4.1: Exact $h$-PMD

**Lemma A.2** (Triple Point Descent Lemma for $h$-PMD). *Let* $h \in \mathbb{N} \setminus \{0\}$. *Initialized at* $\pi_0 \in \mathrm{ri}(\Pi)$, *the iterates of* $h$-step greedy PMD remain in $\mathrm{ri}(\Pi)$ and satisfy the following inequality for any $k \in \mathbb{N}$ *and any policy* $\pi \in \Pi$,

$$\eta_k \mathcal{T}^\pi \mathcal{T}^{h-1} V^{\pi_k} \leq \eta_k \mathcal{T}^{\pi_{k+1}} \mathcal{T}^{h-1} V^{\pi_k} + D_\phi(\pi, \pi_k) - D_\phi(\pi, \pi_{k+1}) - D_\phi(\pi_{k+1}, \pi_k)$$

*Proof.* Recall the definition of the $h$-step greedy action value function denoted by $Q_h^\pi : \mathcal{S} \times \mathcal{A} \to \mathbb{R}$ in (2). Then, recall that for each state $s \in \mathcal{S}$ and every $k \in \mathbb{N}$, we have

$$\langle Q_h^{\pi_k}(s, \cdot), \pi_s \rangle = (\mathcal{T}^\pi \mathcal{T}^{h-1} V^{\pi_k})(s). \tag{9}$$

Applying the three point descent lemma (see Lemma 6 in [53] with $\phi(\pi) = \langle Q_h^{\pi_k}(s, \cdot), \pi \rangle$ for any $\pi \in \Pi$ and $\mathcal{C} = \Delta(\mathcal{A})$ with the notation therein), we obtain for any policy $\pi \in \Pi$,

$$-\eta_k \langle Q_d^{\pi_k}(s, \cdot), \pi_s^{k+1} \rangle + D_\phi(\pi_s^{k+1}, \pi_s^k) \leq -\eta_k \langle Q_d^{\pi_k}(s, \cdot), \pi_s \rangle + D_\phi(\pi_s, \pi_s^k) - D_\phi(\pi_s, \pi_s^{k+1}).$$

We obtain the desired result by rearranging the above last inequality and using (9). $\square$

**Lemma A.3.** *Let* $h \in \mathbb{N} \setminus \{0\}$. *Initialized at* $\pi_0 \in \mathrm{ri}(\Pi)$, *the iterates of* $h$-step greedy PMD satisfy *for every* $k \in \mathbb{N}$,

$$\mathcal{T}^{\pi_{k+1}} \mathcal{T}^{h-1} V^{\pi_k} \leq V^{\pi_{k+1}} + \frac{\gamma}{1-\gamma} c_k \mathbf{e},$$

*where* $(c_k)$ *is the sequence of positive integers introduced in Theorem 4.1 such that* $\frac{\tilde{d}_k}{\eta_k} \leq c_k$ *and* $\tilde{d}_k = \|\min_{\pi \in \mathcal{G}_h(V^{\pi_k})} D_\phi(\pi, \pi_k)\|_\infty$.

*Proof.* Let $k \in \mathbb{N}$. Recall that any policy $\pi \in \mathcal{G}_h(V^{\pi_k})$ satisfies $\mathcal{T}^\pi \mathcal{T}^{h-1} V^{\pi_k} = \mathcal{T}^h V^{\pi_k}$. Using Lemma A.2 for $\pi \in \mathcal{G}_h(V^{\pi_k})$ together with the previous identity and the nonnegativity of the Bregman divergence implies that for any for any policy $\pi \in \mathcal{G}_h(V^{\pi_k})$,

$$\mathcal{T}^h V^{\pi_k} \le \mathcal{T}^{\pi_{k+1}} \mathcal{T}^{h-1} V^{\pi_k} + \frac{\tilde{d}_k}{\eta_k} \mathbf{e} \le \mathcal{T}^{\pi_{k+1}} \mathcal{T}^{h-1} V^{\pi_k} + c_k \mathbf{e}, \tag{10}$$

where the last inequality follows from the assumption on the stepsizes $\eta_k$. We now prove by recurrence that for every $n \in \mathbb{N}$,

$$\mathcal{T}^{\pi_{k+1}} \mathcal{T}^{h-1} V^{\pi_k} \le (\mathcal{T}^{\pi_{k+1}})^{n+1} \mathcal{T}^{h-1} V^{\pi_k} + \sum_{i=1}^{n} \gamma^i c_k \mathbf{e}, \tag{11}$$

with $\sum_{i=1}^{n} \gamma^i = 0$ for $n = 0$. The base case for $n = 0$ is immediate. We now suppose that the result holds for $n$ and we show that it remains true for $n + 1$. We have

$$\mathcal{T}^{\pi_{k+1}} \mathcal{T}^{h-1} V^{\pi_k} = \mathcal{T}^{\pi_{k+1}} \mathcal{T}^{h-1} \mathcal{T}^{\pi_k} V^{\pi_k} \qquad\qquad (\mathcal{T}^{\pi_k} V^{\pi_k} = V^{\pi_k})$$

$$\le \mathcal{T}^{\pi_{k+1}} \mathcal{T}^h V^{\pi_k} \quad (\mathcal{T}^{\pi_k} V \le \mathcal{T} V \text{ for any } V \in \mathbb{R}^S \text{ and monotonicity of } \mathcal{T}^{\pi_{k+1}})$$

$$\le \mathcal{T}^{\pi_{k+1}} (\mathcal{T}^{\pi_{k+1}} \mathcal{T}^{h-1} V^{\pi_k} + c_k \mathbf{e}) \qquad (\text{using (10) and monotonocity of } \mathcal{T}^{\pi_{k+1}})$$

$$= (\mathcal{T}^{\pi_{k+1}})^2 \mathcal{T}^{h-1} V^{\pi_k} + \gamma c_k \mathbf{e}$$
$$(\mathcal{T}^\pi (V + c\mathbf{e}) = \mathcal{T}^\pi V + \gamma c \mathbf{e} \text{ for any } V \in \mathbb{R}^S, c \in \mathbb{R})$$

$$\le \mathcal{T}^{\pi_{k+1}} \left( (\mathcal{T}^{\pi_{k+1}})^{n+1} \mathcal{T}^{h-1} V^{\pi_k} + \sum_{i=1}^{n} \gamma^i c_k \mathbf{e} \right) + \gamma c_k \mathbf{e}$$
$$(\text{using recurrence hypothesis and monotonicity of } \mathcal{T}^{\pi_{k+1}})$$

$$= (\mathcal{T}^{\pi_{k+1}})^{n+2} \mathcal{T}^{h-1} V^{\pi_k} + \sum_{i=1}^{n} \gamma^{i+1} c_k \mathbf{e} + \gamma c_k \mathbf{e}$$

$$= (\mathcal{T}^{\pi_{k+1}})^{n+2} \mathcal{T}^{h-1} V^{\pi_k} + \sum_{i=1}^{n+1} \gamma^i c_k \mathbf{e}, \tag{12}$$

which concludes the recurrence proof of (11) for every $n \in \mathbb{N}$. To conclude, we take the limit in (11) and use the fact that for any $V \in \mathbb{R}^S$, the sequence $((\mathcal{T}^{\pi_{k+1}})^n V)_n$ converges to the fixed point $V^{\pi_{k+1}}$ of the contractive operator $\mathcal{T}^{\pi_{k+1}}$. Hence, we obtain

$$\mathcal{T}^{\pi_{k+1}} \mathcal{T}^{h-1} V^{\pi_k} \le V^{\pi_{k+1}} + \sum_{i=1}^{+\infty} \gamma^i c_k \mathbf{e} = V^{\pi_{k+1}} + \frac{\gamma}{1-\gamma} c_k \mathbf{e}. \tag{13}$$

This concludes the proof. $\qquad\square$

*Remark* A.4. We note that although in our work the stepsise $\eta_k$ is chosen to be the same for all states for simplicity, the same result holds for a stepsize that is different for each state as long as $\eta_k(s) \ge \frac{1}{c_k} D_\phi(\pi_s, \pi_s^k)$ for each $s \in \mathcal{S}$. The same is true for all results depending on this lemma (or the inexact variant).

## A.2 Proof of Theorem 4.1

As a warmup, we start by reporting a convergence result for the standard PMD algorithm, recently established in [20]. Using the contraction properties of the Bellman operators, it can be shown that the suboptimality gap $\|V^{\pi_k} - V^*\|_\infty$ for PI iterates converges to zero at a linear rate with a contraction factor $\gamma$, regardless of the underlying MDP. This instance-independent convergence rate was generalized to PMD in [20]. Before stating this result, we report a useful three-point descent lemma to quantify the improvement of the PMD policy resulting from the proximal step compared to an arbitrary policy.

**Lemma A.5** (Three Point Descent Lemma). *Using the PMD update rule, we have for every $k \in \mathbb{N}$ and every $\pi \in \Pi$,*

$$-\eta_k \mathcal{T}^{\pi_{k+1}} V^{\pi_k} \le -\eta_k \mathcal{T}^\pi V^{\pi_k} - D_\phi(\pi_{k+1}, \pi_k) + D_\phi(\pi, \pi_k) - D_\phi(\pi, \pi_{k+1}). \tag{14}$$

This lemma results from an application of Lemma 6 in [53] which is a variant of Lemma 3.2 in [10]. This same result was also used in [20].

**Theorem A.6** (Exact PMD, [20]). *Let $(c_k)$ be a sequence of positive reals and let the stepsize $\eta_k$ in (PMD) satisfy $\eta_k \geq \frac{1}{c_k}\|\min_{\pi \in \mathcal{G}(V^{\pi_k})} D_\phi(\pi, \pi_k)\|_\infty$ where the minimum is computed component-wise. Initialized at $\pi_0 \in \mathrm{ri}(\Pi)$, the iterates $(\pi_k)$ of (PMD) satisfy for every $k \in \mathbb{N}$,*

$$\|V^\star - V^{\pi_k}\|_\infty \leq \gamma^k \left( \|V^\star - V^{\pi_0}\|_\infty + \sum_{t=1}^{k} \gamma^{-t} c_{t-1} \right).$$

We provide here a complete proof of this result as a warmup for our upcoming main result. Although equivalent, notice that our proof is more compact compared to the one provided in [20] (see section 6 and Lemma A.1 to A.3 therein) thanks to the use of our convenient notations, and further highlights the relationship between PMD and PI through the explicit use of Bellman operators.

*Proof.* Using Lemma A.5 with a policy $\pi \in \mathcal{G}(V^{\pi_k})$ for which $\mathcal{T}^\pi V^{\pi_k} = \mathcal{T} V^{\pi_k}$, we have

$$\eta_k \mathcal{T} V^{\pi_k} \leq \eta_k \mathcal{T}^{\pi_{k+1}} V^{\pi_k} - D_\phi(\pi_{k+1}, \pi_k) + D_\phi(\pi, \pi_k) - D_\phi(\pi, \pi_{k+1}). \tag{15}$$

Setting $d_k := \|\min_{\pi \in \mathcal{G}(V^{\pi_k})} D_\phi(\pi, \pi_k)\|_\infty$ and recalling that $D_\phi(\pi, \pi_k) \geq 0$ for any policy $\pi \in \Pi$, we obtain

$$\mathcal{T} V^{\pi_k} \leq \mathcal{T}^{\pi_{k+1}} V^{\pi_k} + \frac{d_k}{\eta_k} \mathbf{e}. \tag{16}$$

Notice that for Policy Iteration we have $\mathcal{T} V^{\pi_k} = \mathcal{T}^{\pi_{k+1}} V^{\pi_k}$ i.e. $\pi_{k+1} \in \mathcal{G}(V^{\pi_k})$. Instead, for PMD we have that $\pi_{k+1}$ is approximately greedy (see (16)).

Applying Lemma A.5 again with $\pi = \pi_k$, recalling that $\mathcal{T}^{\pi_k} V^{\pi_k} = V^{\pi_k}$ and that $D_\phi(\pi, \pi') \geq 0$ for any $\pi, \pi' \in \Pi$, we immediately obtain $V^{\pi_k} \leq \mathcal{T}^{\pi_{k+1}} V^{\pi_k}$. Iterating this relation and using the monotonicity of the Bellman operator, we obtain $V^{\pi_k} \leq \mathcal{T}^{\pi_{k+1}} V^{\pi_k} \leq (\mathcal{T}^{\pi_{k+1}})^2 V^{\pi_k} \leq \cdots \leq (\mathcal{T}^{\pi_{k+1}})^n V^{\pi_k} \leq \cdots \leq V^{\pi_{k+1}}$ where the last inequality follows from the fact that $\mathcal{T}^{\pi_{k+1}}$ has a unique fixed point $V^{\pi_{k+1}}$. We are now ready to control the suboptimality gap as follows:

$$
\begin{aligned}
V^\star - V^{\pi_{k+1}} &\leq V^\star - \mathcal{T}^{\pi_{k+1}} V^{\pi_k} \\
&= V^\star - \mathcal{T} V^{\pi_k} + \mathcal{T} V^{\pi_k} - \mathcal{T}^{\pi_{k+1}} V^{\pi_k} \\
&\leq V^\star - \mathcal{T} V^{\pi_k} + \frac{d_k}{\eta_k} \mathbf{e} \\
&\leq V^\star - \mathcal{T} V^{\pi_k} + c_k \mathbf{e},
\end{aligned}
$$

where the first inequality follows from the inequality $V^{\pi_{k+1}} \geq \mathcal{T}^{\pi_{k+1}} V^{\pi_k}$ proved above, the second one stems from (16), and the last one is a consequence of our assumption $d_k/\eta_k \leq c_k$. We conclude the proof by taking the max norm, applying the triangle inequality and using the contractiveness of the Bellman optimality operator $\mathcal{T}$.

□

**Proof of Theorem 4.1**

*Proof.* Similarly to Lemma A.5, we show that for any $\pi \in \mathcal{G}_h(V^{\pi_k})$,

$$\eta_k \mathcal{T}^\pi \mathcal{T}^{h-1} V^{\pi_k} \leq \eta_k \mathcal{T}^{\pi_{k+1}} \mathcal{T}^{h-1} V^{\pi_k} - D_\phi(\pi_{k+1}, \pi_k) + D_\phi(\pi, \pi_k) - D_\phi(\pi, \pi_{k+1}), \tag{17}$$

see Lemma A.2 for a proof. Setting $\tilde{d}_k := \|\min_{\pi \in \mathcal{G}_h(V^{\pi_k})} D_\phi(\pi, \pi_k)\|_\infty$ (note the difference with $d_k$ in the proof of Theorem A.6), and recalling that $\pi \in \mathcal{G}_h(V^{\pi_k})$ implies that $\pi \in \mathcal{G}(\mathcal{T}^{h-1} V^{\pi_k})$, we obtain from (17) an inequality similar to (16),

$$\mathcal{T}^h V^{\pi_k} \leq \mathcal{T}^{\pi_{k+1}} \mathcal{T}^{h-1} V^{\pi_k} + \frac{\tilde{d}_k}{\eta_k} \mathbf{e}. \tag{18}$$

Then, we observe that

$$
\begin{aligned}
\mathcal{T}^{\pi_{k+1}}\mathcal{T}^{h-1}V^{\pi_k} &= \mathcal{T}^{\pi_{k+1}}\mathcal{T}^{h-1}\mathcal{T}^{\pi_k}V^{\pi_k}\\
&\leq \mathcal{T}^{\pi_{k+1}}\mathcal{T}^{h}V^{\pi_k}\\
&\leq \mathcal{T}^{\pi_{k+1}}(\mathcal{T}^{\pi_{k+1}}\mathcal{T}^{h-1}V^{\pi_k} + \frac{\tilde{d}_k}{\eta_k}\mathbf{e})\\
&= (\mathcal{T}^{\pi_{k+1}})^2\mathcal{T}^{h-1}V^{\pi_k} + \gamma\frac{\tilde{d}_k}{\eta_k}\mathbf{e}\\
&\leq (\mathcal{T}^{\pi_{k+1}})^2\mathcal{T}^{h-1}V^{\pi_k} + \gamma c_k\mathbf{e}\,,
\end{aligned}
\tag{19}
$$

where the second inequality follows from using (17) and monotonicity of $\mathcal{T}^{\pi_{k+1}}$ and the last inequality uses the assumption on the stepsizes. Then we show by recursion by iterating (19), taking the limit and using the fact that $\mathcal{T}^{\pi_{k+1}}$ has a unique fixed point $V^{\pi_{k+1}}$, that

$$
\mathcal{T}^{\pi_{k+1}}\mathcal{T}^{h-1}V^{\pi_k} \leq V^{\pi_{k+1}} + \frac{\gamma}{1-\gamma}c_k\mathbf{e}\,,
\tag{20}
$$

see Lemma A.3 for a complete proof. We now control the suboptimality gap as follows,

$$
\begin{aligned}
V^{\star} - V^{\pi_{k+1}} &\leq V^{\star} - \mathcal{T}^{\pi_{k+1}}\mathcal{T}^{h-1}V^{\pi_k} + \frac{\gamma}{1-\gamma}c_k\mathbf{e}\\
&= V^{\star} - \mathcal{T}^{h}V^{\pi_k} + \frac{\gamma}{1-\gamma}c_k\mathbf{e} + \mathcal{T}^{h}V^{\pi_k} - \mathcal{T}^{\pi_{k+1}}\mathcal{T}^{h-1}V^{\pi_k}\\
&\leq V^{\star} - \mathcal{T}^{h}V^{\pi_k} + \frac{1}{1-\gamma}c_k\mathbf{e}\,,
\end{aligned}
\tag{21}
$$

where the first inequality stems from (20) and the last one from reusing (18). Similarly to Theorem A.6, we conclude the proof by taking the max norm, applying the triangle inequality and using the contractiveness of the Bellman optimality operator $h$ times to obtain

$$
\|V^{\star} - V^{\pi_{k+1}}\|_{\infty} \leq \gamma^{h}\|V^{\star} - V^{\pi_k}\|_{\infty} + \frac{1}{1-\gamma}c_k\,.
$$

A recursion gives the desired result. $\qquad\square$

### A.3 Additional Comments About the Stepsizes in Theorem 4.1

**About the cost of computing adaptive stepsizes.** There is typically only one greedy policy at each iteration, and in the case that there is more than one it is possible to just pick any arbitrary greedy policy and the lower bound condition on the step size will still be satisfied. The cost of computing the step size is typically the cost of computing a Bregman divergence between two policies. We elaborate on this in the following using the discussion in [20] p. 7 in 'Computing the step size' paragraph which also applies to our setting. We use the step size condition $\eta_k \geq \frac{1}{c_k}\|\min_{\pi \in G_h(V^{\pi_k})} D_{\phi}(\pi, \pi_k)\|_{\infty}$. We have a minimum in the condition which means we can just pick a single greedy policy $\tilde{\pi} \in G_h(V^{\pi_k})$ (which can easily be computed given a lookahead value function in the exact setting or its estimate in an inexact setting). As for the maximum over the state space, this is because we use the same step size in all the states. This condition can readily be replaced by a state dependent step size $\eta_k(s) \geq \frac{1}{c_k}D_{\phi}(\tilde{\pi}(\cdot|s), \pi_k(\cdot|s))$ which is enough for our results to hold.

**About the condition on the stepsizes.** First, notice that PMD has been analyzed using an increasing stepsize in the exact setting (see e.g. [25, 53, 29]) and this is natural as this corresponds to emulating Policy Iteration. In the present work, notice that we do not require the stepsize to go to infinity and the stepsize does not typically go to infinity under our condition. Recall that we only require the stepsize to satisfy the condition: $\eta_k \geq \frac{1}{c_k}\|\min_{\pi \in \mathcal{G}_h(V^{\pi_k})} D_{\phi}(\pi, \pi_k)\|_{\infty}$. As the policy gets closer to the optimal one it should also get closer to a greedy policy (the optimal policy is greedy with respect to its value function) and the term containing a Bregman divergence will vanish (since the optimal policy is already greedy with respect to its own value function). This effect balances the decreasing $c_k$. Notice that we typically choose $c_k = \gamma^{2h(k+1)}$.

We refer the reader to section D.4 for additional observations regarding stepsizes in experiments.

# B Additional Details and Proofs for Section 5: Inexact $h$-PMD

## B.1 Details and Precise Notation for Lookahead Computation

We describe here our approach to estimate $Q_h^\pi$ in more detail and introduce the notation necessary for the analysis in the following sections. Assume that we are estimating $Q_h^\pi(s,a)$ at a given fixed state-action pair $(s,a) \in \mathcal{S} \times \mathcal{A}$. We define a sequence $(\mathcal{S}_k)$ of sets of visited states as follows: Set $\mathcal{S}_h = \{s\}$ and define for every $k \in [0, h-1]$, for every $\tilde{s} \in \mathcal{S}_{k+1}$ and for ever $\tilde{a} \in \mathcal{A}$ the multiset $\mathcal{S}_k(\tilde{s}, \tilde{a}) := \{s_j' \sim P(\cdot|\tilde{s}, \tilde{a}) : j \in [M_{k+1}]\}$ (including duplicated states) where $M_{k+1}$ is the number of states sampled from each state $\tilde{s} \in \mathcal{S}_{k+1}$, and $\mathcal{S}_k = \bigcup_{\tilde{s} \in \mathcal{S}, \tilde{a} \in \mathcal{A}} \mathcal{S}_k(\tilde{s}, \tilde{a}) \cup \mathcal{S}_{k+1}$ (without duplicated states). The estimation procedure is recursive. First we define the value function $\hat{V}_1$ by $\hat{V}_1(\tilde{s}) := \hat{V}^\pi(\tilde{s})$, $\forall \tilde{s} \in \mathcal{S}_0$, where $\hat{V}^\pi(\tilde{s})$ is an estimate of $V^\pi(\tilde{s})$ computed using $M_0$ rollouts of the policy $\pi$ from state $\tilde{s}$. More precisely, we sample $M_0$ trajectories $(\tau_j(\tilde{s}))_{j \in [M_0]}$ of length $H \geq 1$ rolling out policy $\pi$, i.e. $\tau_j(\tilde{s}) := \{(s_k^j, a_k^j, r(s_k^j, a_k^j))\}_{0 \leq k \leq H-1}$ with $s_0^j = \tilde{s}$ and $a_k^j \sim \pi(\cdot|s_k^j)$. Then the Monte Carlo estimator is computed as $\hat{V}^\pi(\tilde{s}) = \frac{1}{M_0} \sum_{j=1}^{M_0} \sum_{k=0}^{H-1} \gamma^k r(s_k^j, a_k^j)$. For each stage $k \in [1, h]$ for $h \geq 1$, we define the following estimates for every $\tilde{s} \in \mathcal{S}_k, \tilde{a} \in \mathcal{A}$,

$$\hat{Q}_k(\tilde{s}, \tilde{a}) := r(\tilde{s}, \tilde{a}) + \frac{\gamma}{M_k} \sum_{s' \in \mathcal{S}_{k-1}} \hat{V}_k(s'), \quad \hat{V}_{k+1}(\tilde{s}) := \max_{\tilde{a} \in \mathcal{A}} \hat{Q}_k(\tilde{s}, \tilde{a}), \tag{22}$$

The desired estimate $\hat{Q}_h^\pi(s,a)$ is obtained with $k = h$. The recursive procedure described above can be represented as a (partial incomplete) tree where the root node is the state-action pair $(s,a)$ and each sampled action leads to a new state node $s'$ followed by a new state-action node $(s', a')$. To trigger the recursion, we assign estimated values $\hat{V}^\pi(\tilde{s})$ of the true value function $V^\pi(\tilde{s})$ to each one of the leaf state nodes $\tilde{s}$ of this tree, i.e. the state level $h-1$ of this tree which corresponds to the last states visited in the sampled trajectories. The desired estimate $\hat{Q}_h^\pi(s,a)$ is given by the estimate obtained at the root state-action node $(s,a)$. Recalling that $Q_h^\pi(s,a) := r(s,a) + \gamma(PV_h^\pi)(s,a)$ for any $s \in \mathcal{S}, a \in \mathcal{A}$ and $V_h^\pi := \mathcal{T}^{h-1}V^\pi$, the overall estimation procedure consists in using Monte Carlo rollouts to estimate the value function $V^\pi$ before approximating $V_h^\pi := \mathcal{T}^{h-1}V^\pi$ using approximate Bellman operator steps and finally computing the estimate $\hat{Q}_h^\pi(s,a)$ using a sampled version of $Q_h^\pi(s,a) := r(s,a) + \gamma(PV_h^\pi)(s,a)$.

## B.2 Proof of Theorem 5.3: Inexact $h$-PMD

Throughout this section, we suppose that the assumptions of Theorem 5.3 hold.

**Lemma B.1.** *Let $\pi \in \Pi, k \in \mathbb{N}$. Suppose that the estimator $\hat{Q}_h^{\pi_k}$ is such that $\|\hat{Q}_h^{\pi_k} - Q_h^\pi\|_\infty \leq b$ for some positive scalar $b$. Then the approximate $h$-step lookahead value function $\hat{V}_h^{\pi_k}(\pi) := M^\pi \hat{Q}_h^{\pi_k}$ is a $b$-approximation of $\mathcal{T}^\pi \mathcal{T}^{h-1}V^{\pi_k}$, i.e.,*

$$\|\hat{V}_h^{\pi_k}(\pi) - \mathcal{T}^\pi \mathcal{T}^{h-1}V^{\pi_k}\|_\infty \leq b. \tag{23}$$

*Proof.* The Hölder inequality implies that for any policy $\pi \in \Pi$:

$$\|\hat{V}_h^{\pi_k}(\pi) - \mathcal{T}^\pi \mathcal{T}^{h-1}V^{\pi_k}\|_\infty = \max_{s \in \mathcal{S}} |\langle \hat{Q}_h^{\pi_k}(s, \cdot) - Q_h^{\pi_k}(s, \cdot), \pi_s \rangle|$$

$$\leq \max_{s \in \mathcal{S}} \|\hat{Q}_h^{\pi_k}(s, \cdot) - Q_h^{\pi_k}(s, \cdot)\|_\infty \cdot \|\pi_s\|_1 \leq b.$$

$\square$

**Lemma B.2** (Approximate $h$-Step Greedy Three-Point Descent Lemma). *For any policy $\pi \in \Pi$, the iterates $\pi_k$ of approximate $h$-PMD satisfy the following inequality for every $k \geq 0$:*

$$\eta_k \mathcal{T}^\pi \mathcal{T}^{h-1}V^{\pi_k} \leq \eta_k \mathcal{T}^{\pi_{k+1}} \mathcal{T}^{h-1}V^{\pi_k} + D_\phi(\pi, \pi_k) - D_\phi(\pi, \pi_{k+1}) - D_\phi(\pi_{k+1}, \pi_k) + 2\eta_k b\mathbf{e}. \tag{24}$$

*Proof.* We apply again the Three-Point Descent Lemma (see e.g. Lemma A.1 in [20]) to obtain:

$$-\eta_k \langle \hat{Q}_h^{\pi_k}(s, \cdot), \pi_s^{k+1} \rangle + D_\phi(\pi_s^{k+1}, \pi_s^k) \leq -\eta_k \langle \hat{Q}_h^{\pi_k}(s, \cdot), \pi_s \rangle + D_\phi(\pi_s, \pi_s^k) - D_\phi(\pi_s, \pi_s^{k+1}), \tag{25}$$

which by rearranging and expressing in vector form gives:

$$\eta_k \hat{V}_h^{\pi_k}(\pi) \leq \eta_k \hat{V}_h^{\pi_k}(\pi_{k+1}) + D_\phi(\pi, \pi_k) - D_\phi(\pi, \pi_{k+1}) - D_\phi(\pi_{k+1}, \pi_k) \,. \tag{26}$$

Lemma B.1 implies that $\hat{V}_h^{\pi_k}(\pi) \geq \mathcal{T}^\pi \mathcal{T}^{h-1} V^{\pi_k} - b\mathbf{e}$ and $\hat{V}_h^{\pi_k}(\pi_{k+1}) \leq \mathcal{T}^{\pi_{k+1}} \mathcal{T}^{h-1} V^{\pi_k} + b\mathbf{e}$ yielding the lemma. $\qquad\square$

The following lemma is an analogue of (18) in the proof of Theorem 4.1 above, modified for the inexact case.

**Lemma B.3.** *Assuming access at each iteration to an approximate $Q_h$ value function $\hat{Q}_h^{\pi_k}$ with $\|\hat{Q}_h^{\pi_k} - Q_h^\pi\|_\infty \leq b$, the iterates of inexact h-PMD satisfy for every $k \geq 0$:*

$$\mathcal{T}^h V^{\pi_k} \leq \mathcal{T}^{\pi_{k+1}} \mathcal{T}^{h-1} V^{\pi_k} + 2b\mathbf{e} + \frac{\tilde{d}_k}{\eta_k} \mathbf{e} \,. \tag{27}$$

*Proof.* Let $\bar{\pi}$ be any true $h$-step greedy policy $\bar{\pi} \in \mathcal{G}_h(V^{\pi_k})$, and let $\tilde{\pi}$ be any approximate $h$-step greedy policy $\tilde{\pi} \in \operatorname{argmax}_{\pi \in \Pi} \hat{V}_h^{\pi_k}(\pi)$. By definition of $\tilde{\pi}$ and equation (26) we have:

$$\hat{V}_h^{\pi_k}(\bar{\pi}) \leq \hat{V}_h^{\pi_k}(\tilde{\pi}) \leq \hat{V}_h^{\pi_k}(\pi_{k+1}) + \frac{1}{\eta_k} D_\phi(\tilde{\pi}, \pi_k) - \frac{1}{\eta_k} D_\phi(\tilde{\pi}, \pi_{k+1}) - \frac{1}{\eta_k} D_\phi(\pi_{k+1}, \pi_k) \tag{28}$$

$$\leq \hat{V}_h^{\pi_k}(\pi_{k+1}) + \frac{\tilde{d}_k}{\eta_k} \mathbf{e} \,. \tag{29}$$

Finally we have the desired result using Lemma B.1, recalling the argument from the end of Lemma B.2 that $\hat{V}_h^{\pi_k}(\bar{\pi}) \geq \mathcal{T}^{\bar{\pi}} \mathcal{T}^{h-1} V^{\pi_k} - b\mathbf{e} = \mathcal{T}^h V^{\pi_k} - b\mathbf{e}$ and $\hat{V}_h^{\pi_k}(\pi_{k+1}) \leq \mathcal{T}^{\pi_{k+1}} \mathcal{T}^{h-1} V^{\pi_k} + b\mathbf{e}$.

$$\square$$

**Lemma B.4.** *Given a $\hat{Q}_h^{\pi_k}$ with $\|\hat{Q}_h^{\pi_k} - Q_h^\pi\|_\infty \leq b$, let $\tilde{\mathcal{G}}_h = \operatorname{argmax}_{\pi \in \Pi} \hat{V}_h^{\pi_k}(\pi)$ denote the set of approximate h-step greedy policies and $\tilde{d}_k = \|\min_{\tilde{\pi} \in \tilde{\mathcal{G}}_h} D(\tilde{\pi}, \pi_k)\|_\infty$. Then the iterates of inexact h-PMD satisfy the following bound:*

$$\eta_k \mathcal{T}^{\pi_{k+1}} \mathcal{T}^{h-1} V^{\pi_k} \leq \eta_k V^{\pi_{k+1}} + 2\eta_k \frac{\gamma}{1-\gamma} b\mathbf{e} + \frac{\gamma}{1-\gamma} \tilde{d}_k \mathbf{e} \,. \tag{30}$$

*Proof.* Analogously to Lemma A.3 we proceed by induction using Lemma B.3. We write for every $k \geq 0$,

$$\eta_k \mathcal{T}^{\pi_{k+1}} \mathcal{T}^{h-1} V^{\pi_k} \overset{(a)}{\leq} \eta_k \mathcal{T}^{\pi_{k+1}} \mathcal{T}^h V^{\pi_k}$$

$$\overset{(b)}{\leq} \eta_k (\mathcal{T}^{\pi_{k+1}})^2 \mathcal{T}^{h-1} V^{\pi_k} + 2\gamma\eta_k b\mathbf{e} + \gamma\tilde{d}_k \mathbf{e}$$

$$\leq \eta_k (\mathcal{T}^{\pi_{k+1}})^2 \mathcal{T}^h V^{\pi_k} + 2\gamma\eta_k b\mathbf{e} + \gamma\tilde{d}_k \mathbf{e}$$

$$\leq \eta_k (\mathcal{T}^{\pi_{k+1}})^3 \mathcal{T}^{h-1} V^{\pi_k} + 2\gamma^2\eta_k b\mathbf{e} + \gamma^2\tilde{d}_k + 2\gamma\eta_k b\mathbf{e} + \gamma\tilde{d}_k \mathbf{e}$$

$$\leq \ldots$$

$$\leq \eta_k V^{\pi_{k+1}} + 2\eta_k \frac{\gamma}{1-\gamma} b\mathbf{e} + \frac{\gamma}{1-\gamma} \tilde{d}_k \mathbf{e} \,, \tag{31}$$

where (a) follows from the monotonicity of both Bellman operators $\mathcal{T}^{\pi_{k+1}}, \mathcal{T}$ and the fact that $V^{\pi_k} \leq \mathcal{T} V^{\pi_k}$, (b) uses Lemma B.3 and monotonicity of the Bellman operators. The last inequality stems from the fact that for every $V \in \mathbb{R}^S, \lim_{n \to +\infty} (\mathcal{T}^{\pi_{k+1}})^n V = V^{\pi_{k+1}}$ since $V^{\pi_{k+1}}$ is the unique fixed point of the Bellman operator $\mathcal{T}^{\pi_{k+1}}$ which is a $\gamma$-contraction. $\qquad\square$

**End of the proof of Theorem 5.3.** We are now ready to conclude the proof of the theorem. Letting $\tilde{d}_k = \|\min_{\tilde{\pi} \in \tilde{\mathcal{G}}_h} D(\tilde{\pi}, \pi_k)\|_\infty$ as in the theorem, we have:

$$
\begin{aligned}
V^\star - V^{\pi_{k+1}} &\overset{(i)}{\le} V^\star - \mathcal{T}^{\pi_{k+1}} \mathcal{T}^{h-1} V^{\pi_k} + \frac{\gamma}{1-\gamma} \frac{\tilde{d}_k}{\eta_k} \mathbf{e} + 2 \frac{\gamma}{1-\gamma} b\mathbf{e} \\
&= V^\star - \mathcal{T}^h V^{\pi_k} + \mathcal{T}^h V^{\pi_k} - \mathcal{T}^{\pi_{k+1}} \mathcal{T}^{h-1} V^{\pi_k} + \frac{\gamma}{1-\gamma} \frac{\tilde{d}_k}{\eta_k} \mathbf{e} + 2 \frac{\gamma}{1-\gamma} b\mathbf{e} \\
&\overset{(ii)}{\le} V^\star - \mathcal{T}^h V^{\pi_k} + \frac{1}{1-\gamma} \frac{\tilde{d}_k}{\eta_k} \mathbf{e} + \frac{2}{1-\gamma} b\mathbf{e} \, ,
\end{aligned}
$$

where (i) follows from B.3 and (ii) from Lemma B.4. By the triangle inequality and the contraction property of $\mathcal{T}$ we obtain a recursion similar to the previous theorems:

$$
\|V^\star - V^{\pi_{k+1}}\|_\infty \le \gamma^h \|V^\star - V^{\pi_k}\|_\infty + \frac{1}{1-\gamma} \frac{\tilde{d}_k}{\eta_k} + \frac{2}{1-\gamma} b \, .
$$

Unfolding this recursive inequality concludes the proof.

### B.3 Proof of Theorem 5.4

We first recall the notations used in our lookahead value function estimation procedure before providing the proof. The procedure we use is a standard online planning method and our proof follows similar steps to the proof provided for example in the lecture notes [48]. Notice though that we are not approximating the optimal Q function but rather the lookahead value function $Q_h^\pi$ induced by a given policy $\pi \in \Pi$ at any state action pair $(s, a) \in \mathcal{S} \times \mathcal{A}$.

Recall that for any $k \in [2, h-1]$, $\hat{V}_k$ and $\hat{Q}_k$ are defined as follows:

$$
\hat{Q}_k(\tilde{s}, \tilde{a}) = r(\tilde{s}, \tilde{a}) + \gamma \frac{1}{M_k} \sum_{s' \in \mathcal{S}_{k-1}} \hat{V}_k(s') \, , \quad \forall \tilde{s} \in \mathcal{S}_k \, , \tag{32}
$$

$$
\hat{V}_k(\tilde{s}) = \max_{a \in \mathcal{A}} \hat{Q}_{k-1}(\tilde{s}, a) \, , \quad \forall \tilde{s} \in \mathcal{S}_{k-1} \, . \tag{33}
$$

At the bottom of the recursion, recall that $\hat{V}_1$ is defined for every $\tilde{s} \in \mathcal{S}_0$ by

$$
\hat{V}_1(\tilde{s}) = \hat{V}^\pi(\tilde{s}) = \frac{1}{M_0} \sum_{j=1}^{M_0} \sum_{k=0}^{H-1} \gamma^k r(s_k^j, a_k^j) \, ,
$$

using $M_0$ trajectories $(\tau_j(\tilde{s}))_{j \in [M_0]}$ of length $H \ge 1$ rolling out policy $\pi$, i.e. $\tau_j(\tilde{s}) := \{(s_k^j, a_k^j, r(s_k^j, a_k^j))\}_{0 \le k \le H-1}$ with $s_0^j = \tilde{s}$ and $a_k^j \sim \pi(\cdot|s_k^j)$.

Note that the cardinality of each set $\mathcal{S}_k$ is evidently upper bounded by $S$ but also by $A^{h-k} \prod_{i \in [k+1, h]} M_i$ as the number of states in each set grows by a factor of at most $AM_k$ at step $k$. In this section we prove some concentration bounds on the above estimators, which will be useful for the rest of the analysis.

**Lemma B.5** (Concentration Bounds of Lookahead Value Function Estimators). *Let $\delta \in (0, 1)$. For any state $\tilde{s} \in \mathcal{S}_0$, with probability $1 - \delta$ we have:*

$$
|\hat{V}_1(\tilde{s}) - V^\pi(\tilde{s})| \le C_1^V(\delta) := \frac{\gamma^H}{1-\gamma} + \frac{1}{1-\gamma} \sqrt{\frac{\log(2/\delta)}{M_1}} \, . \tag{34}
$$

*For any $k \in [1, h]$, for state $\tilde{s} \in \mathcal{S}_k$ and for any action $\tilde{a} \in \mathcal{A}$, with probability $1 - \delta$ we have:*

$$
|\hat{Q}_k(\tilde{s}, \tilde{a}) - (J\hat{V}_k)(\tilde{s}, \tilde{a})| \le C_k^Q(\delta) := \frac{\gamma}{1-\gamma} \sqrt{\frac{\log(2/\delta)}{M_k}} \, . \tag{35}
$$

*Finally, for any $k \in [1, h-1]$, for any $\tilde{s} \in \mathcal{S}_k$, also with probability $1 - \delta$ we have:*

$$
|\hat{V}_{k+1}(\tilde{s}) - (\mathcal{T}\hat{V}_k)(\tilde{s})| \le C_{k+1}^V(\delta) := \frac{\gamma}{1-\gamma} \sqrt{\frac{\log(2A/\delta)}{M_k}} \, . \tag{36}
$$

*Proof.* The proof of this result is standard and relies on Hoeffding's inequality. We provide a proof for completeness. Let $\Pr$ be the probability measure induced by the interaction of the planner and the MDP simulator on an adequate probability space. We denote by $\mathbb{E}$ the corresponding expectation.

First we show the bound for the value function estimate $\hat{V}_1$. This estimate is biased due to the truncation of the rollouts at a fixed horizon $H$, which is the source of the first error term in the bound. For any state $\tilde{s} \in \mathcal{S}_0$, the bias in $\hat{V}_1(\tilde{s})$ is given by

$$|\mathbb{E}\left[\hat{V}_1(\tilde{s})\right] - V^\pi(\tilde{s})| = \mathbb{E}\left[\sum_{t=H}^\infty \gamma^t r(s_t, a_t)|s_0 = \tilde{s}, a_t \sim \pi(\cdot|s_t)\right] \leq \frac{\gamma^H}{1-\gamma}. \tag{37}$$

The second term is due to Hoeffding's inequality: each trajectory is i.i.d, and the estimator is bounded with probability 1 by $\frac{1}{1-\gamma}$. Therefore $\hat{V}_1(\tilde{s})$ concentrates around its mean, and with probability $1-\delta$ we have:

$$|\hat{V}_1(\tilde{s}) - \mathbb{E}\left[\hat{V}_1(\tilde{s})\right]| \leq \frac{1}{1-\gamma}\sqrt{\frac{\log(2/\delta)}{M_1}}. \tag{38}$$

Now we prove the bound for the $\hat{Q}_k$ function. Assume $k \in [1, h]$ is arbitrary, and $\tilde{s} \in \mathcal{S}_k$. For any action $\tilde{a} \in \mathcal{A}$, we can bound the error of $\hat{Q}_k$ as follows:

$$|\hat{Q}_k(\tilde{s}, \tilde{a}) - (J\hat{V}_k)(\tilde{s}, \tilde{a})| = \gamma\left|\frac{1}{M_k}\sum_{i=1}^{M_k}\hat{V}_k(s_i') - (P\hat{V}_k)(\tilde{s}, \tilde{a})\right| \leq \frac{\gamma}{1-\gamma}\sqrt{\frac{\log(2/\delta)}{M_k}}. \tag{39}$$

Finally we bound the $\hat{V}_{k+1}$ function for $k > 1$. Assume $k \in [1, h-1]$ is arbitrary, and $\tilde{s} \in \mathcal{S}_k$. With probability at least $1-\delta$ we have:

$$|\hat{V}_{k+1}(\tilde{s}) - (\mathcal{T}\hat{V}_k)(\tilde{s})| \leq \frac{\gamma}{1-\gamma}\sqrt{\frac{\log(2A/\delta)}{M_k}}. \tag{40}$$

To show this last result, we use a standard union bound. For every $t \geq 0$, we have:

$$\begin{aligned}
&\Pr(|\hat{V}_{k+1}(\tilde{s}) - \mathcal{T}\hat{V}_k(\tilde{s})| \geq t)\\
&= \Pr(|\max_{a'\in\mathcal{A}}\hat{Q}_k(\tilde{s}, a') - \max_{a'\in\mathcal{A}}(JV_k)(\tilde{s}, a')| \geq t)\\
&\leq \Pr(\max_{a'\in\mathcal{A}}\hat{Q}_k(\tilde{s}, a') - \max_{a'\in\mathcal{A}}(JV_k)(\tilde{s}, a') \geq t) + \Pr(\max_{a'\in\mathcal{A}}(JV_k)(\tilde{s}, a') - \max_{a'\in\mathcal{A}}\hat{Q}_k(\tilde{s}, a') \geq t)\\
&\leq \Pr(\max_{a'\in\mathcal{A}}\left[\hat{Q}_k(\tilde{s}, a') - (JV_k)(\tilde{s}, a')\right] \geq t) + \Pr(\max_{a'\in\mathcal{A}}\left[(JV_k)(\tilde{s}, a') - \hat{Q}_k(\tilde{s}, a')\right] \geq t)\\
&\leq \sum_{\tilde{a}\in\mathcal{A}}\Pr(\hat{Q}_k(\tilde{s}, \tilde{a}) - (J\hat{V}_k)(\tilde{s}, \tilde{a}) \geq t) + \sum_{\tilde{a}\in\mathcal{A}}\Pr((J\hat{V}_k)(\tilde{s}, \tilde{a}) - \hat{Q}_k(\tilde{s}, \tilde{a}) \geq t)\\
&\leq 2A\exp(-2\frac{(1-\gamma)^2}{\gamma^2}t^2 M_k). \tag{41}
\end{aligned}$$

$\square$

### B.3.1 Estimation Error Bound for $\hat{Q}_h$

We start by defining some useful Bellman operators. The Bellman operator $J : \mathbb{R}^S \to \mathbb{R}^{S\times A}$ and its sampled version $\hat{J} : \mathbb{R}^S \to \mathbb{R}^{S\times A}$ are defined for every $V \in \mathbb{R}^S$ and any $(s, a) \in \mathcal{S} \times \mathcal{A}$ by

$$JV(s, a) := r(s, a) + \gamma PV(s, a), \tag{42}$$

$$\hat{J}V(s, a) := r(s, a) + \frac{\gamma}{m}\sum_{s'\in\mathcal{C}(s,a)}V(s'), \tag{43}$$

where $\mathcal{C}(s, a)$ is a set of $m$ states sampled i.i.d from the distribution $P(\cdot|s, a)$.

**Lemma B.6.** *The operators $T$ and $\hat{T}$ satisfy for every $U, V \in \mathbb{R}^S, s \in \mathcal{S}, a \in \mathcal{A}$,*

$$|JU(s,a) - JV(s,a)| \leq \gamma\|U - V\|_\infty\,, \tag{44}$$

$$|\hat{J}U(s,a) - \hat{J}V(s,a)| \leq \gamma \max_{s' \in \mathcal{C}(s,a)} |U(s) - V(s)|\,. \tag{45}$$

*Proof.* The proof is immediate and is hence omitted. $\qquad\square$

We now show how to bound the error of $\hat{Q}_k$ for any $k \in [2, h]$. We can bound the total error on $\hat{Q}_k$ by decomposing the error into parts, accounting individually for the contribution of each stage in the estimation procedure. We will make use of the operators $\hat{J}$ as defined above and we will index them by the sets $\mathcal{S}_k$ ($k \in [1, h]$) for clarity. We use the notation $\|\cdot\|_{\mathcal{X}}$ for the infinity norm over any finite set $\mathcal{X}$. Recall that $Q_k^\pi = r + \gamma P\mathcal{T}^{k-1}V^\pi = J\mathcal{T}^{k-1}V^\pi$ for every $k \in [1, h]$. The following decomposition holds for any $k \in [2, h]$ using the aforementioned contraction properties:

$$\max_{a \in \mathcal{A}}\|\hat{Q}_k(\cdot, a) - Q_k^\pi(\cdot, a)\|_{\mathcal{S}_k}$$

$$= \max_{a \in \mathcal{A}}\|(\hat{J}_{\mathcal{S}_{k-1}}\hat{V}_k)(\cdot, a) - (J\mathcal{T}^{k-1}V^\pi)(\cdot, a)\|_{\mathcal{S}_k}$$

$$\leq \max_{a \in \mathcal{A}}\|(\hat{J}_{\mathcal{S}_{k-1}}\hat{V}_k)(\cdot, a) - (\hat{J}_{\mathcal{S}_{k-1}}\mathcal{T}^{k-1}V^\pi)(\cdot, a)\|_{\mathcal{S}_k} + \max_{a \in \mathcal{A}}\|(\hat{J}_{\mathcal{S}_{k-1}}\mathcal{T}^{k-1}V^\pi)(\cdot, a) - (J\mathcal{T}^{k-1}V^\pi)(\cdot, a)\|_{\mathcal{S}_k}$$

$$\leq \gamma\|\hat{V}_k - \mathcal{T}^{k-1}V^\pi\|_{\mathcal{S}_{k-1}} + \max_{a \in \mathcal{A}}\|(\hat{J}_{\mathcal{S}_{k-1}}\mathcal{T}^{k-1}V^\pi)(\cdot, a) - (J\mathcal{T}^{k-1}V^\pi)(\cdot, a)\|_{\mathcal{S}_k}$$

$$= \gamma\|M\hat{Q}_{k-1} - MJ\mathcal{T}^{k-2}V^\pi\|_{\mathcal{S}_{k-1}} + \max_{a \in \mathcal{A}}\|(\hat{J}_{\mathcal{S}_{k-1}}\mathcal{T}^{k-1}V^\pi)(\cdot, a) - (J\mathcal{T}^{k-1}V^\pi)(\cdot, a)\|_{\mathcal{S}_k}$$

$$\leq \gamma \max_{a \in \mathcal{A}}\|\hat{Q}_{k-1}(\cdot, a) - (J\mathcal{T}^{k-2}V^\pi(\cdot, a)\|_{\mathcal{S}_{k-1}} + \max_{a \in \mathcal{A}}\|(\hat{J}_{\mathcal{S}_{k-1}}\mathcal{T}^{k-1}V^\pi)(\cdot, a) - (J\mathcal{T}^{k-1}V^\pi)(\cdot, a)\|_{\mathcal{S}_k}$$

$$= \gamma \max_{a \in \mathcal{A}}\|\hat{Q}_{k-1}(\cdot, a) - Q_{k-1}^\pi(\cdot, a)\|_{\mathcal{S}_{k-1}} + \max_{a \in \mathcal{A}}\|(\hat{J}_{\mathcal{S}_{k-1}}\mathcal{T}^{k-1}V^\pi)(\cdot, a) - (J\mathcal{T}^{k-1}V^\pi)(\cdot, a)\|_{\mathcal{S}_k} \tag{46}$$

which, after defining the terms $\Delta_k = \max_{a \in \mathcal{A}}\|\hat{Q}_k(\cdot, a) - Q_k^\pi(\cdot, a)\|_{\mathcal{S}_k}$ and

$\mathcal{E}_k = \max_{a \in \mathcal{A}}\|(\hat{J}_{\mathcal{S}_{k-1}}\mathcal{T}^{k-1}V^\pi)(\cdot, a) - (J\mathcal{T}^{k-1}V^\pi)(\cdot, a)\|_{\mathcal{S}_k}$ yields the following recursion:

$$\Delta_k \leq \gamma\Delta_{k-1} + \mathcal{E}_k\,, \tag{47}$$

and unrolling this recursion leads to the following bound on $\Delta_h$:

$$\Delta_h \leq \gamma^{h-1}\Delta_1 + \sum_{i=2}^{h} \gamma^{h-i}\mathcal{E}_i\,. \tag{48}$$

We begin by bounding $\Delta_1$ by partially applying the recursion above:

$$\Delta_1 = \max_{a \in \mathcal{A}}\|\hat{Q}_1(\cdot, a) - JV^\pi(\cdot, a)\|_{\mathcal{S}_1} = \max_{a \in \mathcal{A}}\|\hat{J}_{\mathcal{S}_0}\hat{V}_1(\cdot, a) - JV^\pi(\cdot, a)\|_{\mathcal{S}_1} \leq \gamma\|\hat{V}_1 - V^\pi\|_{\mathcal{S}_0}\,, \tag{49}$$

which from Lemma B.5 we know to be bounded with probability at most $1 - \delta|\mathcal{S}_0|$ by:

$$\Delta_1 \leq \gamma\|\hat{V}_1 - V^\pi\|_{\mathcal{S}_0} \leq \frac{\gamma^{H+1}}{1-\gamma} + \frac{\gamma}{1-\gamma}\sqrt{\frac{\log(2|\mathcal{S}_0|/\delta)}{M_0}}\,. \tag{50}$$

Finally, each term $\mathcal{E}_j$ for $j \leq k$ can be bounded using the following lemma below. The proof relies on a union bound over the states reached during the construction of the search tree, see the lemma towards the end of [48] for a detailed explanation and a proof.

**Lemma B.7.** *Let $n(j) = |\mathcal{S}_j|$ for $j \in [k]$ and any $k \in [h]$, let $S_1$ be the root state of the tree search and for each $1 < i \leq n(j)$ let $S_i \in \mathcal{S}_j$ be the state reached at the $i - 1^{th}$ simulator call such that $\mathcal{S}_j = \{S_i\}_{i=1}^{n(j)}$. For $0 < \delta \leq 1$ with probability $1 - An(j)\delta$ for any $1 \leq i \leq n(j)$ we have:*

$$\max_{a \in \mathcal{A}}|(\hat{J}_{\mathcal{S}_{j-1}}\mathcal{T}^{k-1}V^\pi)(S_i, a) - (J\mathcal{T}^{k-1}V^\pi)(S_i, a)| \leq \frac{\gamma}{1-\gamma}\sqrt{\frac{\log(2/\delta)}{M_j}}\,. \tag{51}$$

Combining this result with (50) and the unrolled recursion gives the final concentration bound with probability at least $1 - \delta_V - \delta_J$ for $\delta_V, \delta_J \in (0, 1)$,

$$\max_{a \in \mathcal{A}} \|\hat{Q}_k(\cdot, a) - Q_k^\pi(\cdot, a)\|_{\mathcal{S}_k} \leq \frac{\gamma^{H+k}}{1-\gamma} + \frac{\gamma^k}{1-\gamma}\sqrt{\frac{\log(2|\mathcal{S}_0|/\delta_V)}{M_0}} + \frac{\gamma^2(1-\gamma^{k-1})}{(1-\gamma)^2}\sqrt{\frac{\log(2A|\mathcal{S}_0|(k-1)/\delta_J)}{M}} \tag{52}$$

by a union bound over $j \leq k$, assuming $M_j = M$ for all $1 \leq j \leq h - 1$.

### B.3.2 End of the Proof of Theorem 5.4

For any state-action pair $(s, a) \in \mathcal{S} \times \mathcal{A}$, define the set $|\mathcal{S}_h| = \{s\}$ and the sets $|\mathcal{S}_k|$ accordingly for $0 \leq k \leq h - 1$. Here we set $M_t := M > 0$, $\forall t \in [1, h]$. Assuming the worst case growth of $|S_k| \leq |S_{k+1}| A M_{k+1}$, we can bound $|\mathcal{S}_0|$ deterministically as $|\mathcal{S}_0| \leq A^h M^h$. Using the bound in the previous section, it is possible to bound the error of $\hat{Q}_h(s, a)$ as follows:

$$|\hat{Q}_h(s, a) - Q_h^\pi(s, a)| \leq \max_{a' \in \mathcal{A}} \|\hat{Q}_h(\cdot, a') - Q_h^\pi(\cdot, a')\|_{\mathcal{S}_h}$$

$$\leq \frac{\gamma^{H+h}}{1-\gamma} + \frac{\gamma^h}{1-\gamma}\sqrt{\frac{\log(2|\mathcal{S}_0|/\delta_V)}{M_0}} + \frac{\gamma^2(1-\gamma^{h-1})}{(1-\gamma)^2}\sqrt{\frac{\log(2A|\mathcal{S}_0|h/\delta_J)}{M}}, \tag{53}$$

with probability at least $1 - \delta_V - \delta_J$.

Assume now we want to estimate the value function $\hat{Q}(s, a)$ for all state action pairs $(s, a)$ in some set $\mathcal{C} \subseteq \mathcal{S} \times \mathcal{A}$: we want to achieve $\max_{(s,a) \in \mathcal{C}} |\hat{Q}(s, a) - Q(s, a)| \leq b$ for some positive accuracy $b$ with probability at least $1 - \delta_V - \delta_J$. This can be achieved by choosing the following parameters:

$$\forall 1 \leq j \leq h, M_j := M > \frac{9\gamma^4(1-\gamma^{h-1})^2}{(1-\gamma)^4 b^2}\log(2hA|\mathcal{S}_0||\mathcal{C}|/\delta_J), \tag{54}$$

$$M_0 > \frac{9\gamma^{2h}}{(1-\gamma)^2 b^2}\log(2|\mathcal{S}_0||\mathcal{C}|/\delta_V), \tag{55}$$

$$H > \frac{1}{1-\gamma}\log\left(\frac{3\gamma^h}{b(1-\gamma)}\right). \tag{56}$$

Picking $\mathcal{C} = \mathcal{S} \times \mathcal{A}$ gives $\|\hat{Q}_h - Q_h^\pi\|_\infty \leq b$. If estimates for $\{V_k\}_{k=1}^h$ are reused across state action pairs, and the components of each value function are only computed at most once for every state $s \in S$, then the overall number of samples is of the order of $KM_0H|\mathcal{S}_0| + Kh|\mathcal{S}_0| \cdot |\mathcal{C}|M$. In particular, with $\mathcal{C} = \mathcal{S} \times \mathcal{A}$, the number of samples becomes of the order of $KM_0HS + KhS \cdot AM$ where we reuse estimators (which are in this case computed for each state-action pair $(s, a) \in \mathcal{S} \times \mathcal{A}$. Let $K > \frac{1}{h(1-\gamma)}\log(\frac{4}{\epsilon(1-\gamma)(1-\gamma^h)})$ as in Theorem 5.4. Choosing $\delta_V = \delta_J = \delta/2$ for some $\delta \in (0, 1)$ and $b = \frac{\epsilon(1-\gamma)(1-\gamma^h)}{4}$ yields the following bound with probability at least $1 - \delta$ (by Theorem 5.3):

$$\|V^\star - V^{\pi_k}\|_\infty \leq \gamma^{hk}\left(\|V^\star - V^{\pi_0}\|_\infty + \frac{1-\gamma^{hk}}{(1-\gamma)(1-\gamma^h)}\right) + \frac{\epsilon}{2}. \tag{57}$$

To conclude, note that when $k = K$, we have:

$$\gamma^{hK}\left(\|V^\star - V^{\pi_0}\|_\infty + \frac{1-\gamma^{hK}}{(1-\gamma)(1-\gamma^h)}\right) \leq \gamma^{hK}\frac{2(1-\gamma^{hK})}{(1-\gamma)(1-\gamma^h)} \leq \frac{\epsilon(1-\gamma^{hK})}{2} \leq \frac{\epsilon}{2}. \tag{58}$$

## C Additional Details and Proofs for Section 6: $h$-PMD with Function Approximation

To incorporate function approximation to $h$-PMD, the overall approach is as follows: compute an estimate of the lookahead value function by fitting a linear estimator using least squares, then perform an update step as in inexact PMD, taking advantage of the same error propagation analysis which is

agnostic to the source of errors in the value function. However a problem that arises is in storing the new policy, since a classic tabular storage method would require $\mathcal{O}(S)$ memory, which we would like to avoid to scale to large state spaces. A solution to this problem will be presented later on in this section.

Since we are using lookahead in our $h$-PMD algorithm, we modify the targets in our least squares regression procedure in order to approximate $Q_h^\pi$ rather than $Q^\pi$ as in the standard PMD algorithm (with $h = 1$).

## C.1  Inexact $h$-PMD with Linear Function Approximation

Here we provide a detailed description of the $h$-PMD algorithm using function approximation to estimate the $Q_h$-function. Firstly we consider the lookahead depth $h$ as a fixed input to the algorithm. We assume that the underlying MDP has finite state and action spaces $\mathcal{S}$ and $\mathcal{A}$, with transition kernel $P : \mathcal{S} \times \mathcal{A} \to \Delta(\mathcal{A})$, reward vector $R \in [0,1]^{S \times A}$ and discount factor $\gamma$. We also assume we are given a feature map $\psi : \mathcal{S} \times \mathcal{A} \to \mathbb{R}^d$, also written as $\Psi \in \mathbb{R}^{SA \times d}$.

We adapt the inexact $h$-PMD algorithm from section 5 as follows:

1. Take as input any policy $\pi_0 \in \text{rint}(\Pi)$, for example one that is uniform over actions in all states.

2. Initialize our memory containing $\pi_0$, and an empty list $\Theta$ which will hold the parameters of our value function estimates computed at each iteration.

3. At each iteration $k = 0, \ldots, K$, perform the following steps:

   (a) Compute the targets $\hat{R}(z) := \hat{Q}_h(z)$ for all $z \in \mathcal{C}$ using the procedure described in section 5.1, using $\pi_k$ as a policy.

   (b) Compute $\theta_k := \text{argmin}_{\theta \in \mathbb{R}^d} \sum_{z \in \mathcal{C}} \rho(z)(\psi(z)^\top \theta - \hat{R}(z))^2$ and append it to the list $\Theta$.

   (c) Using $\Psi\theta_k$ as an estimate for $Q_h^{\pi_k}$, compute $\pi_{k+1}$ using the usual update from inexact $h$-PMD. This step can be merged with step 3a, the details are described below.

Computing the full policy $\pi_k$ for every state at every iteration would require $\Omega(S)$ operations, although the number of samples to the environment still depends only on $d$. If this is not a limitation the policy update step can be performed as in inexact $h$-PMD simply by using $\Psi\theta_k$ as an estimate for $Q_h^{\pi_k}$, as described in step 3c. Otherwise, we describe a method below to compute $\pi_k$ on demand when it is needed in the computation of the targets in step 3a, eliminating step 3c entirely.

1. The policy $\pi_0$ is already given for all states.

2. For any $k > 0$ given $\pi_k$ we can compute $\pi_{k+1}$ at any state $s$ using the following:

   (a) Compute any greedy policy $\tilde{\pi}(\cdot|s)$ using $\Psi\theta_k$, for example $\tilde{\pi}(a'|s) \propto \mathbb{1}\left\{ a' \in \text{argmax}_{a \in \mathcal{A}} \left\{ \langle \psi(s,a)^\top \theta_k \right\} \right\}$

   (b) Compute $\eta_k := \frac{1}{\gamma^{2h(k+1)}} D_\phi(\tilde{\pi}(\cdot|s), \pi^k(\cdot|s))$

   (c) Finally, compute $\pi_{k+1}(\cdot|s) \in \text{argmax}_{\pi \in \Pi} \{\eta_k \langle (\Psi\theta_k)_s, \pi(\cdot|s) \rangle - D_\phi(\pi(\cdot|s), \pi_k(\cdot|s)) \}$

*Remark* C.1.  This procedure can be used to compute $\pi_{k+1}(\cdot|s)$ "on demand" for any $k > 0$ and $s \in \mathcal{S}$ by recursively computing $\pi_k(\cdot|s')$ at any states $s'$ required for the computation of $\Psi\theta_k$ (since $\pi_0$ is given). The memory cost requirement for this method is of the order of $\mathcal{O}(KA \min\{|\mathcal{S}_0|H, S\})$ in total which is always less than computing the full tabular policy at each iteration.

*Remark* C.2.  Above, $(\Psi\theta_k)_s$ denotes a vector in $\mathbb{R}^A$ that can be computed in $\mathcal{O}(Ad)$ operations: each component in $a$ is equal to $\psi(s,a)^\top \theta_k$ which can be computed in $\mathcal{O}(d)$ operations. Also, the stepsize $\eta_k$ is chosen to be equal to its lower bound in the analysis of 5.3, with the value of $c_k$ chosen to be equal to $\gamma^{2h(k+1)}$, as it is needed for linear convergence. We also note that by memorizing policies as they are computed, we manage to perform just as many policy updates as are needed by the algorithm.

## C.2 Discussion of Assumption 6.2

We make two preliminary comments before discussing the relevance of our assumption and alternatives to relax it below:

- First, notice that we do not require $Q_h^\pi$ to be represented as a linear function. Our assumption is an **approximate** action value function approximation.

- When the lookahead depth is $h = 1$, we recover the standard PMD algorithm. Notice that in this case our assumption is standard in the RL literature for linear function approximation.

Now, using our notations, notice that for any policy $\pi \in \Pi$, $Q_h^\pi = r + \gamma P \mathcal{T}^{h-1} V^\pi = r + \gamma P \mathcal{T}^{h-1} M^\pi Q^\pi = J \mathcal{T}^{h-1} M^\pi Q^\pi$. We have two alternatives:

- Directly approximate $Q_h^\pi$ as we do in our assumption. In that case the targets used in our regression procedure are estimates $\hat{Q}_h$ of $Q_h^\pi$. Note also that this approach is meaningful from a practical point as we may directly use a neural network to approximate the lookahead Q-function which is used in our $h$-PMD algorithm rather than approximating the Q-function first and then trying to approximate from it the lookahead Q-function $Q_h^\pi$.

- Approximate $Q^\pi$ with linear function approximation and use our propagation analysis (see section 5.1 and proof in appendix B.3.1, B.3.2) based on the link between $Q_h^\pi$ and $Q^\pi$ highlighted in the formula above. This approach which propagates the function approximation error on $Q^\pi$ to $Q_h^\pi$ constitutes an alternative to Assumption 6.2 on $Q_h^\pi$ by an assumption on $Q^\pi$. We discuss it further in the rest of this section.

In the following exposition we provide more details regarding estimating $Q^\pi$ using linear function approximation and relaxing assumption 6.2 using the second approach discussed above.

Let Assumption 6.1 hold (as in Theorem 6.3). Instead of assumption 6.2, we introduce the following assumption on the Q-function $Q^\pi$ rather than the $h$-step lookahead Q-function $Q_h^\pi$.

**Assumption C.3.** $\exists \epsilon > 0, \forall \pi \in \Pi, \inf_{\theta \in \mathbb{R}^d} \|Q^\pi - \Psi\theta\| \leq \epsilon$.

We modify the $h$-PMD with function approximation algorithm as follows: instead of directly approximating $Q_h^\pi$ using linear function approximation, we first use linear function approximation to approximate $Q^\pi$ from which we construct an estimate of $Q_h^\pi$ similarly to our method for inexact $h$-PMD. This requires us to replace the regression targets with simpler ones, which are Monte Carlo estimates of $Q^\pi$ instead of $Q_h^\pi$. Specifically, these new targets are defined as:

$$R_{M_0}(z) := \frac{1}{M_0} \sum_{i=1}^{M_0} \sum_{t=0}^{H-1} \gamma^t r(s_t^{(i)}, a_t^{(i)}),$$

where $(s_t^{(i)}, a_t^{(i)})$ are state action pairs sampled according to the policy $\pi$, $z = (s, a) \in \mathcal{C} \subseteq \mathcal{S} \times \mathcal{A}$ and $(s_0^{(i)}, a_0^{(i)}) = z$.

For any policy $\pi$ we can use the same function approximation procedure as described in our paper (see Appendix C.1) to yield a $\hat{\theta} \in \mathbb{R}^d$ with $\|Q^\pi - \Psi\hat{\theta}\|_\infty \leq \epsilon_0$, (for some value of $\epsilon_0$ to be defined below) by simply plugging in $h = 1$. Note that this particular case was previously analysed in [48] (Lecture 8). We reuse an intermediate result therein (equation (7)) which guarantees the bound:

$$\|Q^\pi - \Psi\hat{\theta}\|_\infty \leq \|\epsilon_\pi\|_\infty (1 + \sqrt{d}) + \sqrt{d} \left( \frac{\gamma^H}{1-\gamma} + \frac{1}{1-\gamma} \sqrt{\frac{\log(2|\mathcal{C}|/\delta)}{2M_0}} \right) =: \epsilon_0(\delta) \qquad (59)$$

with probability at least $1 - \delta$ for some $0 < \delta < 1$.

Now we reuse most of our estimation procedure in Section 5.1, simply by changing the definition of $\hat{V}_1$ to $\hat{V}_1(s) := \sum_{a \in \mathcal{A}} \pi(a|s)(\Psi\hat{\theta})_{s,a}$ for all $s \in \mathcal{S}_0$. This evidently retains the guarantee that $|\hat{V}_1(s) - V^\pi(s)| \leq \epsilon_0(\delta)$ with probability at least $1 - \delta$ for each $s \in \mathcal{S}_0$. Finally, using the same error propagation analysis as in Appendix B.3.1, replacing $C_1^V(\frac{\delta_1^V}{|\mathcal{S}_0|})$ with $\epsilon_0(\delta_1^V)$ we obtain the following

bound for any individual state action pair $(s, a)$:

$$|\hat{Q}_h^\pi(s,a) - Q_h^\pi(s,a)| \leq \frac{\gamma}{1-\gamma}\sqrt{\frac{\log(2/\delta_h^Q)}{M_h}} + \gamma^2\frac{1-\gamma^{h-1}}{(1-\gamma)^2}\sqrt{\frac{\log(2(h-1)Z/\delta_t^V)}{M}} + \gamma^h\epsilon_0(\delta_1^V)$$

(60)

with probability at least $1 - \delta_1^V - \delta_t^V - \delta_h^Q$.

Finally, the update rule as described in Appendix C.1 is replaced by $\pi_{k+1}(\cdot|s) \in \mathrm{argmax}_{\pi_s \in \Delta(\mathcal{A})}(\eta_k\langle \hat{Q}_h^{\pi_k}(s,\cdot), \pi_s\rangle - D_\phi(\pi_s, \pi_s^k))$. Note that we only compute $\pi_k$ in the states where we need it to compute $\pi_{k+1}$ as described in Appendix C.1.

Unfortunately, it is not clear how to directly obtain a result similar to Theorem 6.3 from this approach (controlling $\|V^\star - V^{\pi_k}\|_\infty$ at each iteration, i.e. uniformly over all states) beyond the bound that we provide above. A naive union bound over the state action space would result in a bound with a logarithmic dependence on the size of the state space, unlike Theorem 6.3 which results in a bound that is completely independent of the size of the state space.

We would like to add an additional comment regarding possible ways to relax Assumption 6.2 even for the standard case where $h = 1$. Recently, Mei et al. [30] proposed an interesting alternative approach relying on order-based conditions (instead of approximation error based ones) to analyze policy gradient methods with linear function approximation. We would like to highlight though that their work only applies to the bandit setting to the best of our knowledge. The extension to MDPs is far from being straightforward and would require some specific investigation. It would be interesting to see if one can relax Assumption 6.2 using similar ideas.

### C.3 Convergence Analysis of Inexact $h$-PMD using Function Approximation

Our analysis follows the same lines as the proof in [27] under our assumptions from section 6.

We compute $\hat{Q}_h(z)$ for a subset $\mathcal{C}$ of state action pairs using the vanilla Monte Carlo Planning algorithm presented in section 5. These will be our targets for estimating the value function using least squares. In order to control the extrapolation of these estimates to the full $Q_h^\pi$ function, we will need the following theorem.

**Theorem C.4** ([22]). *Assume $\mathcal{Z}$ is finite and $\psi : \mathcal{Z} \to \mathbb{R}^d$ is such that the underlying matrix $\Psi \in \mathbb{R}^{|\mathcal{Z}| \times d}$ is of rank $d$. Then there exist $\mathcal{C} \subseteq \mathcal{Z}$ and a distribution $\rho : \mathcal{C} \to [0,1]$ (i.e. $\sum_{z \in \mathcal{C}} \rho(z) = 1$) satisfying the following conditions:*

1. $|\mathcal{C}| \leq d(d+1)/2$,

2. $\sup_{z \in \mathcal{Z}} \|\psi(z)\|_{\mathcal{G}_\rho^{-1}} \leq \sqrt{d}$ where $\mathcal{G}_\rho := \sum_{z \in \mathcal{C}} \rho(z)\psi(z)\psi(z)^\top$.

Let $\mathcal{C} \subseteq \mathcal{S} \times \mathcal{A}$, $\rho : \mathcal{C} \to [0,1]$ be as described in the Kiefer-Wolfowitz theorem. Define $\hat{\theta}$ as follows:

$$\hat{\theta} \in \mathrm{argmin}_{\theta \in \mathbb{R}^d} \sum_{z \in \mathcal{C}} \rho(z)(\phi(z)^\top \theta - \hat{R}_m(z))^2.$$

(61)

Assuming that $\mathcal{G}_\rho$ is nonsingular, a closed form solution for the above least squares problem is given by $\hat{\theta} = \mathcal{G}_\rho^{-1}\sum_{z \in \mathcal{C}}\rho(z)\hat{R}_m(z)\phi(z)$. This will be used to estimate our lookahead value function as $\Psi\hat{\theta}$. We can control the extrapolation error of this estimate as follows:

**Theorem C.5.** *Let $\delta \in (0,1)$ and let $\theta \in \mathbb{R}^d$ such that $\|Q_h^\pi - \Phi\theta\|_\infty \leq \epsilon_\pi$. Assume that we estimate the targets $\hat{Q}_h$ using the Monte Carlo planning procedure described in section 5.1 with parameters, $M$ and $M_0$. There exists a subset $\mathcal{C} \subseteq \mathcal{Z}$ and a distribution $\rho : \mathcal{C} \to \mathbb{R}_{\geq 0}$ for which, with probability at least $1 - \delta$, the least squares estimator $\hat{\theta}$ satisfies:*

$$\|Q_h^\pi - \Phi\hat{\theta}\|_\infty \leq \epsilon_\pi(1 + \sqrt{d}) + \epsilon_Q\sqrt{d},$$

(62)

*where*

$$\epsilon_Q := \frac{\gamma^{H+h}}{1-\gamma} + \frac{\gamma^h}{1-\gamma}\sqrt{\frac{\log(4Zd^2/\delta)}{M_0}} + \frac{\gamma^2(1-\gamma^{h-1})}{(1-\gamma)^2}\sqrt{\frac{\log(4hZd^2/\delta)}{M}},$$

(63)

*and $Z := A^h M_h M^{h-1}$.*

*Proof.* The proof of this theorem is identical to the one in [27], up to using our new targets and error propagation analysis instead of the typical function approximation targets for $Q$-function estimation (without lookahead). The subset $\mathcal{C} \subseteq \mathcal{S} \times \mathcal{A}$ and distribution $\rho : \mathcal{C} \to [0, 1]$ are given by the Kiefer-Wolfowitz theorem stated above. We suppose we have access to such a set. Note that this assumption is key to controlling the extrapolation error of a least squares estimate $\Psi\hat{\theta}$. $\qquad\qquad\square$

We have now fully described how to estimate the lookahead value function $Q_h^\pi$ using a number of trajectories that does not depend on the size of the state space. In the previous section, we described how to use this estimate in an $h$-PMD style update, while maintaining a memory size that does not depend on $S$. Therefore we have an $h$-PMD algorithm that does not use memory, computation or number of samples depending on $S$.

## D    Further Simulations

### D.1    Simulations for $h$-PMD with Euclidean Regularization

Here we provide details of an additional experiment running $h$-PMD with Euclidean regularization. As in section 7, experiments were run on the *bsuite* DeepSea environment [34] with $\gamma = 0.99$ and a grid size of 64x64. The figures are identical in structure to the ones in 7, 32 runs were performed for each value of $h$ in 1, 5, 10, 15, and 20. In Fig. 3, we report the number of samples to the environment at each iteration for the same runs. When compared to the figures showing the convergence, it is clearly seen that running the algorithm with higher values of lookahead results in convergence using fewer overall samples.

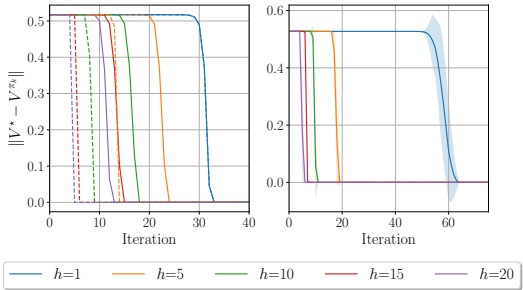

Figure 2: Suboptimality value function gap for $h$-PMD using Euclidean Bregman divergence. in the exact (left) and inexact (right) settings. (Right) We performed 32 runs for each value of $h$, the mean is shown as a solid line and the standard deviation as a shaded area.

### D.2    Sample Complexity Comparison for Different Lookahead Depths

We have recorded the sample complexity of our algorithm for different values of $h$ in Figure 3. Notice that the number of samples used at each iteration increases with $h$. However, notice that this increase is small: $h$-PMD with $h = 20$ requires only 1.5 times more samples per iteration than $h = 1$ as shown in Figure 3. Furthermore, increasing $h$ greatly improves the convergence rate of the algorithm as described by our theory, which results in a much smaller number of iterations required until convergence (for example $h = 20$ converges in at least 10 times fewer iterations than $h = 1$ (see figure 1 (right) in section 7). Overall, the algorithm is more sample efficient for higher values of $h$.

### D.3    Total Running Time Comparison for Different Lookahead Depths

We provide a plot accounting for running time instead of the number of iterations (see Figure 4). Observe that the overall performance is very similar to the same plot with the number of iterations on the x-axis.

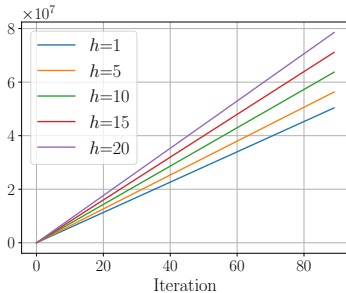

Figure 3: Samples used by inexact $h$-PMD at each iteration. When compared with the first figure, it is clear to see the algorithm needs less samples to converge with higher values of $h$. Since far less iterations are needed with higher values of lookahead, the benefit of higher convergence rate greatly outweighs the additional cost per iteration.

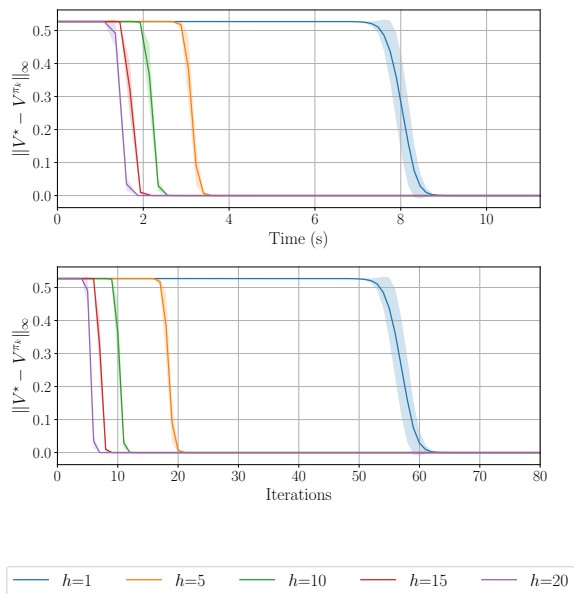

Figure 4: Value function gap against running time. Note that the algorithms with higher values for $h$ still converge faster in terms of runtime.

### D.4 About the Stepsizes in Simulations

In our experiments we use the exact same stepsizes described in our theory. We also observe in our simulations that the stepsize does not go to infinity but rather decreases at some point and/or stabilizes. This is a consequence of the adaptivity of the stepsize discussed above. We also note that the simulations reported in [20] (appendix D p. 18) also validate the fact that the stepsizes do not go to infinity (see the right plots therein for the stepsizes). We also observe a similar behavior for the stepsizes in our experiments (see Figure 5).

### D.5 Larger lookahead depths

In order to practically investigate the tradeoff of increasing the lookahead depth $h$, we performed additional experiments where $h$ was increased to 100 in the DeepSea *bsuite* environment. The plots in Figure 6 show that increasing $h$ past the values in our initial experiments still results in better performance.

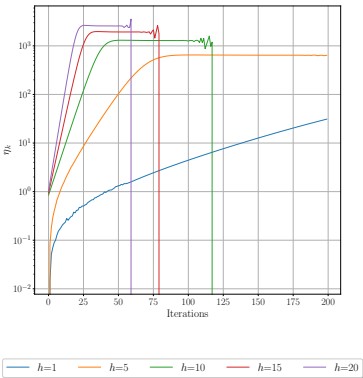

Figure 5: Behaviour of our adaptive stepsize $\eta_k$ at each iteration of the algorithm for different values of $h$. Note that the stepsize does not diverge towards infinity, instead it usually vanishes after a certain number of iterations.

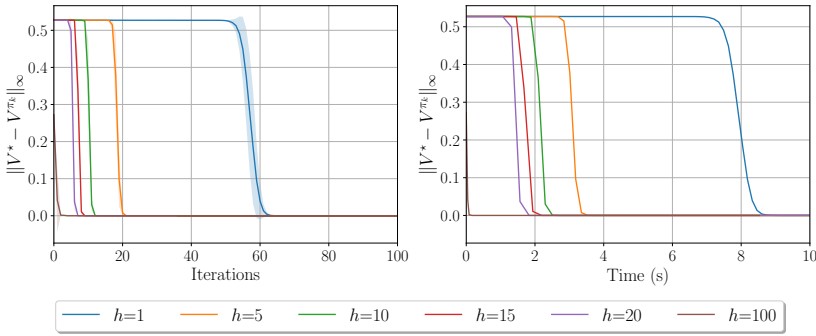

Figure 6: Value function gap plotted against number of iterations (left) and against running time (right) of $h$-PMD in the DeepSea *bsuite* environment. Note that increasing the lookahead depth seems to result in better performance in terms of runtime and number of iterations.

### D.6    Experiments with MCTS-based Lookahead Estimation

We have performed additional experiments in an even more challenging and practical setting, on two environments from the *bsuite* [34] set: DeepSea and Catch. We have implemented our PMD algorithm with a Monte Carlo Tree Search (MCTS) algorithm (which we call PMD-MCTS) to compute the lookahead Q-function. We used DeepMind's MCTS implementation (MCTX)[4] in order to run this algorithm on any deterministic gym style environments implemented in JAX (we use gymnax [26] to implement these environments). Note that any tree search algorithm implementation can be used to estimate the lookahead function and plugged into our PMD method. See figures below. Note that this training procedure is more practical: we do not evaluate the value function at each state-action pair anymore at each time step, we only evaluate the value function (and so only update the policy) in states that we encounter using the current policy. This significantly relaxes the generative model assumption. Note that in this more challenging setting the best performance is not obtained for higher values of $h$: intermediate values of $h$ perform better, illustrating the tradeoff in choosing the depth $h$.

---

[4]https://github.com/google-deepmind/mctx

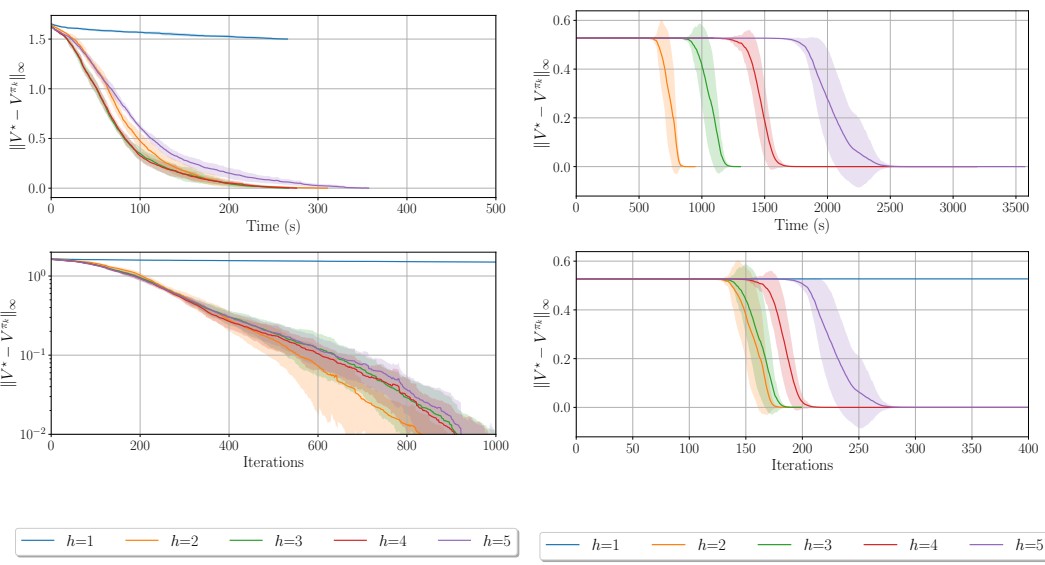

Figure 7: Training curves of PMD-MCTS for the *bsuite* Catch (left) and DeepSea (right) environments. Note that the case of $h = 1$ fails to converge in any reasonable number of iterations in both environments.

## D.7 Continuous Control Experiments

Finally, we performed experiments in classic continuous control environments in order to investigate the performance of $h$-PMD when applied to MDPs with large/infinite state spaces. For this continuous state space setting, we implemented a fully online variant of $h$-PMD based off of DeepMind's MuZero [40] algorithm. Experiments show that increasing the lookahead depth can lead to better performance up to a certain point, and the tradeoff seems to be environment-specific.

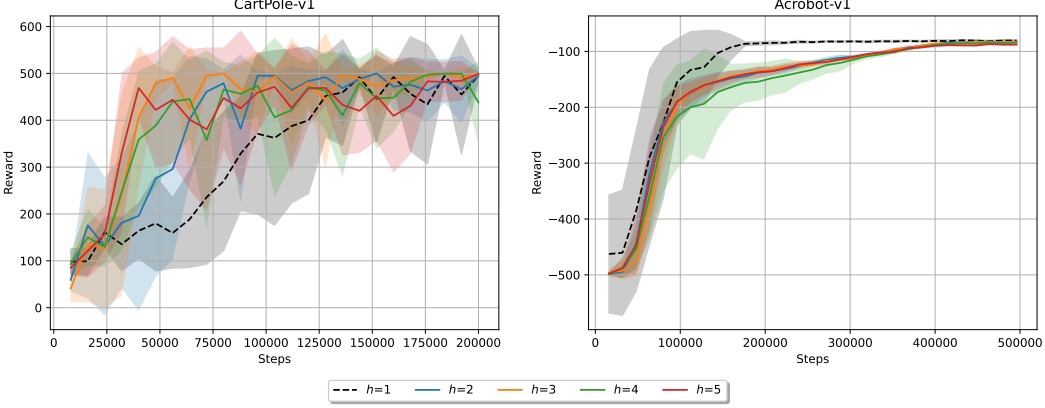

Figure 8: Performance of $h$-PMD in two of OpenAI's *gym* [8] environments, CartPole-v1 (left) and Acrobot-v1 (right). In both figures the reward is plotted against the total number of environment steps evaluated, which can be understood as the number of samples used. Note that higher values of $h$ do lead to better performance for the CartPole-v1 environment, but not necessarily for the Acrobot-v1 environment.

## D.8 Additional experiments

We conclude with results from a larger set of experiments designed to verify the robustness of our experimental results to different environment parameters. We report the performance of the $h$-PMD algorithm in the inexact setting in two different environments, DeepSea from *bsuite* [34], and the

classic Gridworld environment. The effect of $\gamma$ is as expected: higher values of $\gamma$ result in slower convergence, lower values of $\gamma$ yield faster convergence. The size of the environment ("chain length" for DeepSea and number of columns/rows for Gridworld) directly affects the number of iterations until convergence predictably: larger environments take more iterations to solve. Note that in all cases the effect of $h$ is as expected, i.e. higher $h$ leads to faster convergence.

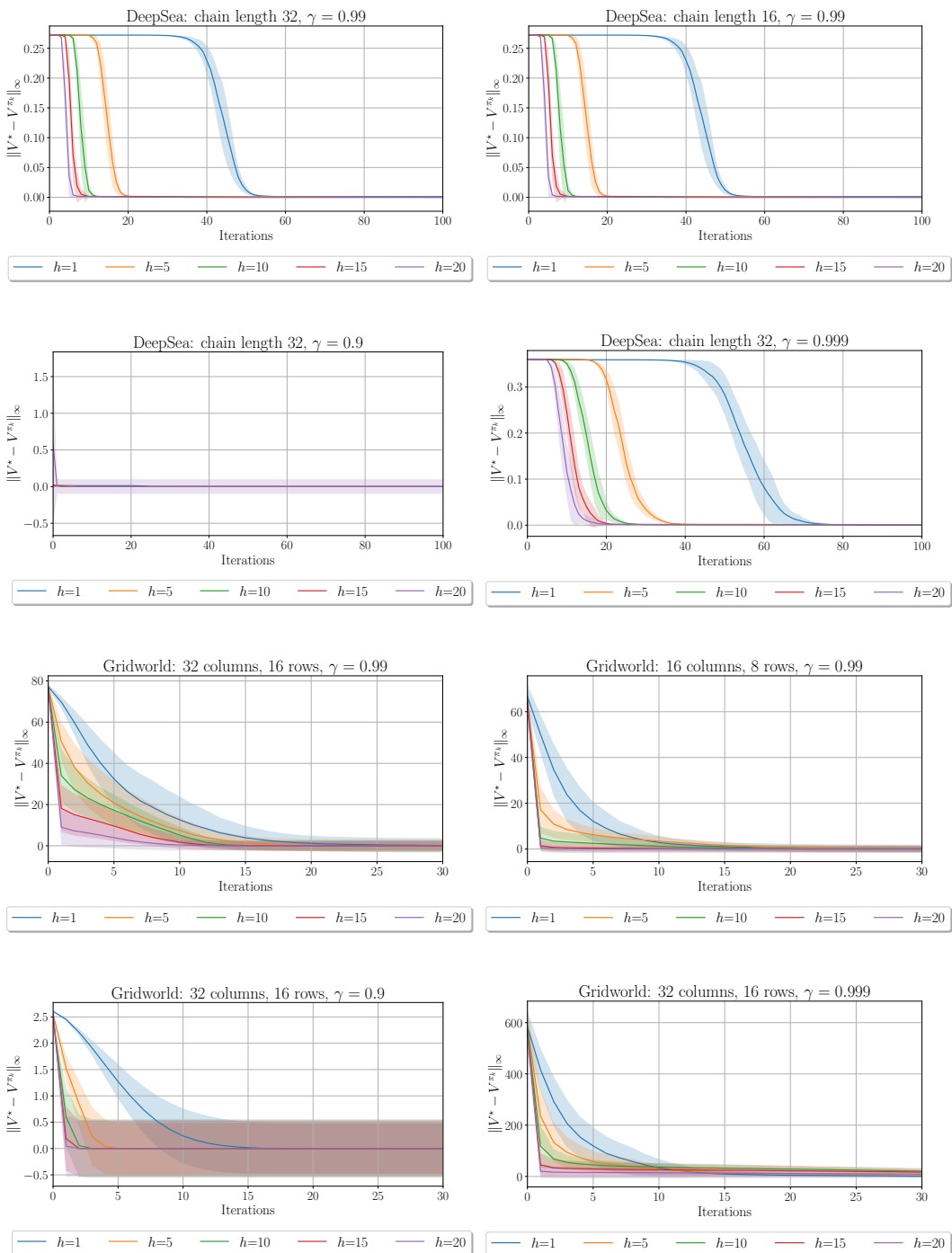

