# OpenReview forum: "Policy Mirror Descent with Lookahead"
_NeurIPS.cc/2024/Conference — NeurIPS 2024 poster_

### Official Review · Reviewer_QSY1 · 2024-06-23

**Soundness:** 3
**Presentation:** 3
**Contribution:** 3
**Rating:** 6
**Confidence:** 4

**Summary:**

The paper studies policy mirror descent (PMD) where the policy improvement step is modified to include an action-value with $h$-step lookahead (contains $h-1$ applications of the Bellman optimality operator on the value of the policy from the previous iteration). In the exact tabular setting where the value functions can be computed exactly, this leads to an improvement of the linear convergence rate of standard PMD from $\gamma$ to $\gamma^h$. In the inexact tabular setting where the the lookahead Q-functions are estimated through Monte-Carlo rollouts, this gives a sample-complexity for finding a $\varepsilon$-optimal policy of $\tilde{O}(1/\varepsilon^2(1-\gamma)^7)$, improving by a factor of $1/(1-\gamma)$ with respect to prior work. Finally, they extend their results beyond the tabular setting to a linear function approximation setting.

**Strengths:**

- The paper is quite well written and easy to follow.
- The paper presents an interesting extension of PMD to include look-ahead leading to faster convergence. The algorithm and results cover different settings from exact tabular to inexact with linear function approximation. Numerical experiments support these claims both in terms of iterations and runtime.

**Weaknesses:**

- The Analysis of Theorem 4.1 closely follows that of [16] while the remaining results (Theorem 5.4 and Theorem 6.3) follow using analyses similar to prior works (except for the estimation of the lookahead Q-function). Nevertheless, I recognise the merit and novelty in combining these with the idea of lookahead from [9]. Note - it would be good to explicitly state where the proof of Theorem 6.3 can be found in the Appendix.
- It seems the only disadvantage of $h$-PMD over $1$-PMD is in the computational cost of computing a more complex value function (lines 212-215) but it is unclear exactly what this cost is (other than what is conveyed in the experiments). It would be nice to explicitly quantify the additional computation (e.g. in the tabular setting, computing the true value function can be done by finding the fixed point of $\mathcal{T}^\pi$ which involves inverting a $S\times S$ matrix, then how much extra computation is required to achieve the value function with lookahead ?). Note: in lines 194-195, the assumption is not only that the value function $V^{\pi_k}$ can be computed exactly but also the value function with lookahead $V^{\pi_k}_h$ right ?
- The rationale behind how to choose $h$ is not clear to me (see also questions below). For example, in Theorem 5.4 (line 270), if the overall sample complexity is $\tilde{O} (\frac{S}{h\varepsilon^2(1-\gamma)^6(1-\gamma^h)^2} + \frac{SA}{\varepsilon^2 (1-\gamma)^7})$, this seems to suggest that we should take $h \rightarrow \infty$. Besides the computational infeasibility of this, if the lookahead Q-functions are estimated using Algorithm 1, then taking $h \rightarrow \infty$ should also result in a sample complexity going to $\infty$, it seems odd that this does not appear in the sample complexity. It would be nice to have an explanation of this.


Typos:
- Line 128: you are missing a “the” in front of squared Euclidean distance.
- Line 564-565, in equation (12), $\mathcal{T}^{\pi_{k+1}}$ should be to the power of $n+2$ not squared.
- Line 344: succesfully -> successfully

**Questions:**

This questions are related to choosing $h$ as discussed in the weaknesses.
- It seems that one step of $h$-PI is equivalent to $h$ steps of value iteration with a policy produced at the end by acting greedily with respect to the last value. Is this the case ? If so, is there a way to choose a good value of $h$ that balances the benefits of VI and PI/PMD ?
- In the simulations, it seems like taking $h$ bigger provides better convergence without the cost of longer runtime. Is this the case for all values of $h$ ? If not, it would be nice to see this reflected in the experiments (i.e. taking much larger values of $h$ which begin to impede on the runtime). If it is true for all values of $h$, is it essentially saying that value iteration is computationally more efficient than PI/PMD in the setting you consider ? And then it would be nice to consider settings where this perhaps is not the case ?

**Limitations:**

Yes

---

> ### Author Rebuttal · Authors · 2024-08-07
>
> We thank the reviewer for their positive feedback on our work and for their useful comments. We are glad the reviewer found the paper "quite well written and easy to follow" and the idea of using lookahead in PMD "interesting".  We address their concerns in the following.
>
> **It would be good to explicitly state where the proof of Theorem 6.3 can be found in the Appendix.**
>
> Thank you for the suggestion, we will add a pointer to the proof (in appendix C.3 p. 26) in the main part.
>
> **It seems the only disadvantage of $h$-PMD over $1$-PMD is in the computational cost of computing a more complex value function (lines 212-215) but it is unclear exactly what this cost is (other than what is conveyed in the experiments). It would be nice to explicitly quantify the additional computation...**
>
> Thank you for the question which we now answer in detail in different settings:
> - Deterministic setting: in this case, the computational overhead of h-PMD over 1-PMD is given by the cost of computing the lookahead action values. These can be computed by performing h steps of VI as the reviewer mentioned. The total cost of this computation can be decomposed into the cost of computing a single value function plus the cost of applying the Bellman optimality operator h times. Overall, the total cost is $O(S^3 + h AS^2)$ which only scales linearly with h compared to h=1 (no lookahead) without any additional dependence on the state nor action space sizes.
>
> - Stochastic setting: in this case, the computational overhead of h-PMD vs PMD is given by the cost of computing the Monte Carlo lookahead action values estimates. Following our procedure in section 5, the induced computational cost for any single state-action pair is that of performing h steps of approximate value iteration which gives a cost of the order of $O(h M S A + H M_0 S)$ where $h$ is the lookahead depth, $M$ is the minibatch size for the sampled transitions for planning, $M_0$ is the number of trajectories for value estimation and $H$ is their horizon length.
> Thanks again for the question, we will add this discussion to the paper.
>
> **Note: in lines 194-195, the assumption is not only that the value function $V^{\pi_k}$ can be computed exactly but also the value function with lookahead $V^{\pi_k}_h$ right ?**
>
> In these lines, notice that we suppose access to a `greedy policy with respect to the value function’ (and not the lookahead one) because greediness refers here to $h$-greedy as defined in l. 149. Another way to formulate it is to suppose access to the lookahead value function from which the 1-step greedy policy can be computed.
>
> **The rationale behind how to choose $h$ is not clear to me (see also questions below). For example, in Theorem 5.4 (line 270), if the overall sample complexity is $\tilde{O} (\frac{S}{h\varepsilon^2(1-\gamma)^6(1-\gamma^h)^2} + \frac{SA}{\varepsilon^2 (1-\gamma)^7})$, this seems to suggest that we should take $h \rightarrow \infty$. Besides the computational infeasibility of this, if the lookahead Q-functions are estimated using Algorithm 1, then taking $h \rightarrow \infty$ should also result in a sample complexity going to $\infty$, it seems odd that this does not appear in the sample complexity. It would be nice to have an explanation of this.**
>
> The number of samples used for lookahead action value function estimation (at each iteration of h-PMD) is of the order of $O(h M S A  + H M_0 S)$ where $h$ is the lookahead depth, $M$ is the minibatch size for the sampled transitions for planning, $M_0$ is the number of trajectories for value estimation and $H$ is their horizon length. Indeed, this sample complexity explodes with a growing lookahead depth $h$. However, our theorem 5.4 shows that we need less and less iterations $K$ when $h$ grows, namely $K > 1/h(1-\gamma) log(4/\epsilon(1-\gamma)(1-\gamma^h))$ (see l. 263). Altogether, the total sample complexity is $O(K (h M S A  + H M_0 S))$ which implies that the $1/h$ in K cancels the multiplicative $h$ in the first term of the overall sample complexity. However, notice that the number of iterations $K$ of h-PMD has to be at least 1 to run our algorithm and this condition translates into a condition on the lookahead depth which needs to be no larger than the effective horizon. Increasing $h$ speeds up the convergence rate of the algorithm which allows us to afford a smaller number of iterations $K$. This number of iterations needs to be lower bounded by 1 though.
>
> **Typos:** Thank you for spotting these, we will make sure to correct them.
>
> **It seems that one step of  $h$-PI is equivalent to $h$ steps of value iteration with a policy produced at the end by acting greedily with respect to the last value. Is this the case ? If so, is there a way to choose a good value of $h$ that balances the benefits of VI and PI/PMD?**
>
> Thank you for this interesting comment and question. Indeed, this is correct in the deterministic (exact) setting. It has been shown in Efroni et al. 2018 [9] (see their Theorem 3) that h-PI enjoys a finite iteration policy convergence guarantee which is monotonically improving with $h$ increasing. In the stochastic setting, we are not aware of any policy convergence result in a finite number of iterations for h-PI, let alone h-PMD. We believe it would be interesting to investigate if we can both guarantee value function convergence as well as policy convergence with a suitable lookahead value.

---

> > ### Comment · Reviewer_QSY1 · 2024-08-08
> >
> > Thank you for your response - these clarify most of the points I had raised. I still wonder about the cost of taking h larger, from the new figure 1 in the pdf attached to the rebuttal, it still seems like using h = 100 leads to convergence in one step without any cost with respect to runtime, which as i mentioned suggests that value iteration is computationally more efficient than PI/PMD in this setting. It would be nice to consider an experiment where this is not the case / where we see a trade-off between the improved convergence and the computational cost of a larger h and in particular where at some value of h the convergence benefit is outweighed by the computational cost.

---

> > > ### Author Response · Authors · 2024-08-09
> > > **Thank you for your response**
> > >
> > > Thank you very much for your response and for your interesting follow up comment.
> > >
> > > This is a good point, we agree. Please see figure 6 p. 29 in the appendix for a setting where large lookahead depth becomes slower and do not perform better. Note that in this more challenging practical setting the best performance is not obtained for higher values of h: intermediate values of h perform better, illustrating the tradeoff in choosing the depth h. We will add a discussion in the main part regarding this interesting point.

---

> > > > ### Comment · Reviewer_QSY1 · 2024-08-12
> > > >
> > > > Thanks for pointing this out, I see that this is the case for DeepSea.

---

> > > > > ### Author Response · Authors · 2024-08-13
> > > > > **Thank you**
> > > > >
> > > > > Thank you for acknowledging our response.

---

### Official Review · Reviewer_vPQx · 2024-07-12

**Soundness:** 3
**Presentation:** 3
**Contribution:** 3
**Rating:** 7
**Confidence:** 2

**Summary:**

The authors proposed a version of the policy mirror descent algorithm that uses a multi-step greedy policy improvement operator. Afterward, the authors showed the theoretical benefits of the proposed method by a better contraction rate $\gamma^h$ instead of $\gamma$-contraction for a usual 1-step greedy policy improvement algorithm. Finally, the inexact version of the algorithm was proposed with an improved sample complexity in the finite MDP setting and, under a linear functional approximation setup, guarantees that are independent of the state space size.

**Strengths:**

- The proposed combination of PMD and $h$-step greedy updates implies interesting theoretical results, such as reducing sample complexity with increasing the planning horizon in a very explicit way;
- Strong result with linear functional approximation that gives an implementable algorithm with a polynomial (in $h$) running time;

**Weaknesses:**

- The experimental part of the paper might be improved by running continuous control experiments.

**Questions:**

- I think the paper may benefit from a theoretical example of h-PI with greedy updates not convergent whereas h-PMD converges.

**Limitations:**

The paper is mostly of theoretical nature and thus does not imply any negative societal impact.

---

> ### Author Rebuttal · Authors · 2024-08-07
>
> We thank the reviewer for their positive assessment of our work and for appreciating our theoretical contributions. We reply to their remaining comments in the following.
>
>
> **The experimental part of the paper might be improved by running continuous control experiments.**
>
> Thank you for the suggestion, we have performed simulations in this setting in the rebuttal phase. Please see Figure 3 in our rebuttal which shows the results of running h-PMD in continuous control settings (CartPole-v1 and Acrobot-v1). We will make sure to include these in our paper.
>
> **I think the paper may benefit from a theoretical example of h-PI with greedy updates not convergent whereas h-PMD converges.**
>
> Thank you for this interesting comment. It has been shown in Efroni et al. 2018 [9] that h-PI converges in the deterministic setting. We have observed that h-PI is unstable in the stochastic setting when the lookahead action values are not very accurately estimated and h-PMD fixes this instability. Such an instability has also been observed in prior work for $h=1$. We have performed simulations to illustrate this in an example in the attached pdf using the same lookahead action values estimation precision (see Figure 2 in our rebuttal). It has been shown in Winnicki and Srikant 2023 [48] that PI with lookahead might converge even in the stochastic setting thanks to using larger lookahead. However, such a result requires a sufficiently large lookahead (see their assumption 1.(b)). Our results for $h$-PMD hold for any lookahead depth value $h$. We believe it would be possible to find an instance where stochastic h-PI does not converge (for a fixed not too large $h$) whereas stochastic h-PMD does: This interesting question remains open.

---

> > ### Comment · Reviewer_vPQx · 2024-08-09
> >
> > I would like to thank the authors for their response. I find the explanations satisfactory and will keep my decision as accept.

---

> > > ### Author Response · Authors · 2024-08-11
> > > **Thank you**
> > >
> > > Thank you very much again for your time, for acknowledging our response and for supporting our paper acceptance.

---

### Official Review · Reviewer_SGix · 2024-07-13

**Soundness:** 3
**Presentation:** 4
**Contribution:** 4
**Rating:** 7
**Confidence:** 2

**Summary:**

The author propose a multi-step greedy approach for Policy Mirror Descent. Combing PMD with multiple greedy policy updates results in a faster $\gamma^{h}$ rate improved the previously thought optimal $\gamma$ rate. Additionally, the authors extend their analysis to the stochastic setting and when using function approximation (with linear function approximation).

**Strengths:**

The idea of combining lookahead to PMD is well-motivated and interesting.  The motivation and presentation of the results are explained well. All results also do not rely distribution mismatch coefficients, which is something that appears quite often in prior work

**Weaknesses:**

It seems that the analysis is done in the functional representation of the policy. It would be good to also discuss how the methods could extended for specific policy parameterization such as softmax policies.
Since the resulting bounds do not rely on concentrability, it would be nice to see experiments where the state space is much larger.

**Questions:**

How well does $h$-PMD perform against PI in the deterministic setting?
Could you provide some intuition on why distribution mismatch coefficients does not appear in the stochastic setting? It's surprising to me since there isn't  any explicit exploration.

**Limitations:**

No concerns.

---

> ### Author Rebuttal · Authors · 2024-08-07
>
> We thank the reviewer for their feedback and for acknowledging the novelty and well-founded motivations of our paper as well as its presentation. We hope that the following discussion fully answers your questions.
>
> **It seems that the analysis is done in the functional representation of the policy. It would be good to also discuss how the methods could be extended for specific policy parameterization such as softmax policies.**
>
> Thank you for your comment. While softmax tabular policies are covered by our analysis, we acknowledge that it would be interesting to consider more general policy parametrization. When introducing (neural network) policy parametrization and considering policy parameters as main variables, we lose part of the structure of the policy optimization problem and we introduce additional non-convexity (on top of the non-convexity of the value function as a function of the policy) into the problem when the objective is seen as a function of the policy parameters. Therefore, we expect to obtain weaker results using a different analysis based on gradient dominance properties of the policy optimization objective rather than contraction arguments.
>
> **Since the resulting bounds do not rely on concentrability, it would be nice to see experiments where the state space is much larger.**
>
> We have performed additional experiments during the rebuttal phase, please see Figure 3 in our rebuttal for experiments showing the feasibility of h-PMD in even continuous state spaces. In these experiments we test our algorithm in continuous control problems (CartPole-v1 and Acrobot-v1). Please note that our algorithm converges in a comparable number of iterations to the variant  without lookahead.
>
> **How well does $h$-PMD perform against PI in the deterministic setting?**
>
> Thank you for this question. In the deterministic setting, h-PMD enjoys a faster $\gamma^h$ convergence rate compared to PI without lookahead in terms of value function gap. h-PMD and h-PI enjoy similar guarantees in terms of value function gap when using our adaptive step sizes for h-PMD. In terms of policy convergence (rather than value function gap convergence), note that PI also enjoys some strong convergence guarantees: it is guaranteed to converge to an optimal policy in a finite number of iterations. As noted in (Efroni et al., 2018, Theorem 3) this result generalizes to $h$-PI as well, yielding a finite iteration complexity guarantee that improves monotonically with $h$. It would be interesting to see if h-PMD can also enjoy such a guarantee in future works, given the relationship between h-PI and h-PMD (see the discussion after Theorem 4.2). That being said about the deterministic setting, note that the true strength of $h$-PMD is in the stochastic setting, where PI is known to be unstable. Please see Figure 2 in our rebuttal which illustrates this in a practical setting.
>
> **Could you provide some intuition on why distribution mismatch coefficients does not appear in the stochastic setting? It's surprising to me since there isn't any explicit exploration.**
>
> From the technical viewpoint, we remove the dependence on distribution mismatch coefficients by avoiding the use of the performance difference lemma (as previously shown by Johnson et al. 2023) and by conducting an analysis which builds on the strong connection between h-PI and h-PMD.
> Notice that the performance error bound in Theorem 5.3 (for inexact h-PMD) features two terms: the first one which stems from the $\gamma^h$ contractiveness of the $h$-step Bellman operator is the same as in the exact setting (Theorem 4.1) and does not involve any distribution mismatch coefficient whereas the second one is a bias term due to lookahead value function estimation that can be made arbitrarily small upon choosing the right minibatch size. In our stochastic setting, we rely on the standard generative model assumption. The (inexact) value function is being computed at every state in the tabular setting, precluding the need for exploration. In this setting, our results are consistent with the work of Johnson et al. 2023 [16] focusing on the particular case of $h=1$. We expect that exploration will play an important role in the online setting where the value function is estimated using online trajectories (i.e. generated according to the current policy at each time step).

---

> > ### Comment · Reviewer_SGix · 2024-08-12
> >
> > I appreciate the authors for their throughout response. My initial questions and concerns have been addressed. I will keep my decision as accept.

---

> > > ### Author Response · Authors · 2024-08-12
> > > **Thank you for your response**
> > >
> > > Thank you very much for your time, for your response and for supporting our paper acceptance.  We are glad that our rebuttal has addressed your questions and concerns.

---

### Official Review · Reviewer_t6qC · 2024-07-14

**Soundness:** 3
**Presentation:** 2
**Contribution:** 2
**Rating:** 4
**Confidence:** 4

**Summary:**

The paper introduces h-PMD, an extension of the Policy Mirror Descent (PMD) algorithm, which incorporates multi-step lookahead to improve policy updates in reinforcement learning. PMD is a general framework that includes several policy gradient methods and relates to advanced algorithms like TRPO and PPO. Recognizing that multi-step greedy policies often outperform their single-step counterparts, the authors propose h-PMD, which integrates multi-step lookahead depth into PMD. This new class of algorithms achieves a faster convergence rate for solving discounted infinite horizon MDPs, under both exact and inexact settings. The paper also extends these results to linear function approximation, demonstrating improved sample complexity that depends on the dimension of the feature map space rather than the state space size.

**Strengths:**

1. The paper introduces a novel extension to the PMD framework by incorporating multi-step lookahead, enhancing the algorithm's performance and convergence rate.

2. The proposed h-PMD algorithm achieves a faster 𝛾^ℎ-linear convergence rate, which is an improvement over standard PMD and Policy Iteration methods.

3. The paper addresses both exact and inexact settings, providing a sample complexity analysis that demonstrates improved performance over previous methods, especially with increasing lookahead depth.

4. By extending h-PMD to linear function approximation, the authors make the algorithm applicable to large state spaces, which is crucial for practical applications in complex environments.

5. The theoretical findings are supported by empirical results from simulations on the DeepSea RL environment, illustrating the benefits of the h-PMD approach

**Weaknesses:**

1. The h-PMD algorithm, while improving convergence rates, also introduces higher computational complexity due to the multi-step lookahead, which can be demanding for large-scale problems.

2. The paper might lack detailed guidance on implementing h-PMD in various practical scenarios, making it challenging for practitioners to adopt and utilize the algorithm effectively.

3. The results are contingent on certain assumptions, such as the availability of multi-step greedy policies and the use of a generative model. These assumptions might limit the generalizability of the findings to all RL problems.

4. The benefits of the h-PMD algorithm are closely tied to the lookahead depth ℎ. Determining the optimal ℎ in practice could be non-trivial and might require extensive experimentation.

5. While the paper demonstrates the theoretical and empirical advantages of h-PMD, it may not provide a comprehensive comparative analysis against a wide range of existing RL algorithms, which would strengthen the case for its superiority.

**Questions:**

1. Can you provide more detailed guidelines on how to implement the h-PMD algorithm in various practical settings, including both exact and inexact scenarios?

2. How does the computational complexity of h-PMD compare to standard PMD and other state-of-the-art RL algorithms like TRPO and PPO in practice? Are there strategies to mitigate the increased computational burden?

3. What methods or heuristics do you recommend for determining the optimal lookahead depth in different environments? How sensitive is the performance of h-PMD to the choice of ℎ?

4. How critical is the assumption of a generative model for the theoretical guarantees provided in the paper? Can h-PMD be effectively applied in settings where a generative model is not available?

5. How well does the h-PMD algorithm with linear function approximation perform in very high-dimensional or continuous state spaces? Have you explored other types of function approximation beyond linear?

6. Can you provide more extensive empirical results across a variety of RL environments to demonstrate the robustness and general applicability of h-PMD?

7. How does h-PMD compare to other advanced RL algorithms like AlphaZero, in terms of performance and computational efficiency? Have you conducted any comparative studies?

8. What are the scalability limits of h-PMD in terms of state and action space sizes? Are there practical cases where h-PMD might not be feasible due to its complexity?

9. Can you elaborate on the trade-offs involved in the inexact setting of h-PMD? How does the estimation of lookahead action values impact the overall performance and convergence rate?

10. Have you applied h-PMD to any real-world RL problems or industrial applications? If so, what were the outcomes and challenges faced?

**Limitations:**

1. The multi-step lookahead in h-PMD increases the computational burden significantly, potentially making it impractical for real-time or resource-constrained applications.

2. The performance of h-PMD is dependent on the chosen lookahead depth ℎ. Finding the optimal depth can be challenging and may require considerable computational resources for tuning.

3. The sample complexity improvements are based on the assumption of a generative model, which may not always be available or practical in many real-world scenarios.

4. While the extension to linear function approximation addresses scalability to large state spaces, the practical implementation and effectiveness of this approach in very high-dimensional or continuous spaces remain uncertain.

5. The empirical validation is limited to the DeepSea RL environment. Broader validation across diverse and more complex environments would be necessary to confirm the general applicability and robustness of the proposed h-PMD algorithm.

6. The paper might not provide sufficient practical implementation details, making it difficult for practitioners to apply the h-PMD algorithm to their specific problems without further guidance and experimentation.

7. Some theoretical results assume exact policy evaluation, which may not be feasible in many practical settings where only approximate evaluations are possible.

---

> ### Author Rebuttal · Authors · 2024-08-07
>
> We thank the reviewer for their time for assessing our work, for their feedback and questions. We reply to each of one of their questions (also covering their weaknesses and limitations comments) in what follows. We will be happy to address any further concern.
>
> **1. Can you provide more detailed guidelines on how to implement the h-PMD algorithm in various practical settings, including both exact and inexact scenarios?**
>
> To support the description of our simulations in section 7, please note that the code for a full implementation of our algorithm is available in the anonymous code repository provided in l. 915-916 (link in section D of the appendix).  We have also implemented a version of our $h$-PMD algorithm with a Monte Carlo Tree Search (MCTS) algorithm to compute the  lookahead Q-function using DeepMind’s MCTS implementation (MCTX) (see section D.5 l. 945-950 in the appendix and the code provided for additional details). All our experiments can be replicated using this code. We will be glad to provide any further details that the reviewer might want to know regarding the implementation.
>
> **2. How does the computational complexity of h-PMD compare to standard PMD and other state-of-the-art RL algorithms like TRPO and PPO in practice? Are there strategies to mitigate the increased computational burden?**
>
> We have compared the performance of our h-PMD algorithm to the standard PMD algorithm corresponding to $h=1$ (for fair comparison) in terms of running time in our simulations (see e.g. Figure 1 right and additional experiments in appendix D.3 Figure 4). We observe that the algorithms with higher values for $h$ still converge faster in terms of runtime, recall that they require less iterations.
>
> Beyond our simulations, exploiting parallel computing is an interesting strategy to further speed up the computation. For instance, this has been used in e.g. [8,15] (to name a few) for implementing tree search methods more efficiently. A version of our algorithm based on Deep Mind’s implementation of MCTS supports such a parallelization which can be useful in very large scale settings.
> In principle, using lookahead value functions can also be used in combination with TRPO and PPO.  Our focus in this work is on showing the potential of lookahead combined with PMD methods (as a general framework encompassing several PG methods as particular cases)  and providing theoretical guarantees supporting our algorithm design. We have performed additional experiments though, please see Figure 3 in our rebuttal, which illustrates the effect of modifying the PPO algorithm to use lookahead value functions.
>
> **3. What methods or heuristics do you recommend for determining the optimal lookahead depth in different environments?...**
>
> The lookahead depth is a hyperparameter of the algorithm and can be tuned similarly to other hyperparameters such as the step size of the algorithm.  Of course, the value would depend on the environment and the structure of the reward at hand. Sparse and delayed reward settings will likely benefit from lookahead with larger depth values. We have performed several simulations with different values of $h$ for each environment setting and the performance can potentially improve drastically with a better lookahead depth value (see section 7 and appendix D for further simulations).
>
> **4. How critical is the assumption of a generative model for the theoretical guarantees provided in the paper? Can h-PMD be effectively applied in settings where a generative model is not available?**
>
> The generative model assumption is important for our theoretical analysis. Relaxing this assumption is an interesting direction for future work. We would like to highlight that (a) this assumption is standard in RL theory and for analyzing PMD (h=1) in particular, see e.g. Xiao 2022 section 5.1, Lan 2023 section 5.1, Johnson et al. 2023, Yuan et al. 2023, Alfano et al. 2023, Li et al. 2023, Zhan et al. 2023 and (b) even under this standard assumption, the analysis requires a careful and involved analysis (see 5.2 and appendix B). Prior work considering lookahead policy iteration algorithms has even focused on the far less challenging deterministic setting assuming that lookahead value functions are accessible (see e.g. [9]).
> In practice, we refer the reader to section D.5, l. 950-956 for a discussion about how to relax it: We only evaluate the value function (and so only update the policy) in states that we encounter using the current policy. We have also implemented our PMD algorithm with a Monte Carlo Tree Search (MCTS) algorithm) to compute the  lookahead Q-function using DeepMind’s MCTS implementation (MCTX) (see section D.5 l. 945-950 in the appendix and the code provided for additional details). The design and analysis of a fully online algorithm is an interesting research question that requires further investigation and that we leave for future work.
>
> **5. How well does the h-PMD algorithm with linear function approximation perform in very high-dimensional or continuous state spaces? Have you explored other types of function approximation beyond linear?**
>
> Thank you for this question. We believe this is an interesting question to address in the future to cover even larger settings as we briefly mention in the conclusion. Using linear function approximation requires designing state-action features which is a delicate task in general as it is notoriously known, especially in high-dimensional spaces. That being said, we have provided a theoretical performance bound for our algorithm in that setting. We are working on using neural networks for approximating the lookahead values to implement an actor-critic style algorithm with lookahead. Please see our experiments on continuous control tasks in Figure 3 of our rebuttal, which illustrates the use of neural net function approximation with h-PMD.
>
> **Please see response to remaining questions (6-10) in the global rebuttal.**

---

> > ### Comment · Area_Chair_Wprb · 2024-08-13
> > **Reminder to address the author's rebuttal**
> >
> > Dear t6qC, please try to respond to the author's feedback today and elaborate if your opinion changed in light of it.

---

### Official Review · Reviewer_Pv8t · 2024-07-16

**Soundness:** 3
**Presentation:** 3
**Contribution:** 2
**Rating:** 4
**Confidence:** 3

**Summary:**

This paper applies the idea of lookahead in policy improvement procedures under the PMD setting. They prove the $\gamma^h$-linear convergence rate. They also propose the inexact version of h-PMD and extend to the function approximation case.

**Strengths:**

The writing of the paper is pretty good. The main idea and the results of the paper are easy to follow.

The paper is also complete. They propose the theoretical analysis in both tabular and function approximation settings.

**Weaknesses:**

Although the paper proved $\gamma^h$-contraction property, it's not out of expectation.

I'm still not completely convinced that $h$-horizon lookahead policy improvement could bring additional benefit in the function approximation case. Even in tabular cases, additional computational effort are required.





Minor problems:

Line 70: bsuite.

**Questions:**

- It's interesting to see that larger lookahead leads to faster training speed based on Figure 1. How about the sample complexities? Could you provide the used samples for each method?
- The meaning of the dotted lines should be clarified in Figure 1 (Left).
- Based on the theoretical result of the paper, does that mean larger $h$ leads to better performance? For Figure 1, what if we have a larger $h$?

**Limitations:**

The paper discusses some potential improvements in future work on the practical algorithm. However, they didn't discuss the limitations of the existing theoretical/practical results.

---

> ### Author Rebuttal · Authors · 2024-08-07
>
> We thank the reviewer for their comments. We are glad that the reviewer finds the writing "pretty good" and the "main idea and the results of the paper easy to follow". We address their remaining concerns in the following. Please let us know if you have any further questions, we will be happy to answer them.
>
> **I'm still not completely convinced that $h$-horizon lookahead policy improvement could bring additional benefit in the function approximation case. Even in tabular cases, additional computational effort are required.**
>
> We also would like to add that multi-step greedy policy improvement contributed to the empirical success of some of the most impressive RL applications including AlphaGo, AlphaZero and MuZero. These practical algorithms have successfully used lookahead (via MCTS) and function approximation jointly in very large state-action space settings although they do not enjoy theoretical value function performance bound guarantees as our work. The benefit of using lookahead policy improvement has also been reported in Winnicki et al. 2021, 2023 [47, 48] for approximate policy iteration with linear function approximation. Using lookahead requires some additional computational effort and we show that it leads to a provably better sample complexity. We illustrate in our simulations how the benefit of using lookahead can greatly outweigh this overhead. Please see also e.g. Figures 6 and 7 in the appendix, in which it is very clear that $h = 1$ (no lookahead) is extremely slow to converge, whereas all versions with lookahead converge much faster.
>
> **Minor problems: Line 70: bsuite.**
>
> `Deep Mind’s bsuite’ (as written in l. 70) is the RL benchmark’s name as introduced in the reference [30]. We will make it in italic to avoid any confusion.
>
> **It's interesting to see that larger lookahead leads to faster training speed based on Figure 1. How about the sample complexities? Could you provide the used samples for each method?**
>
> As we mentioned in l. 325-327 p. 8  in section 7, ‘we also observed in our simulations that h-PMD uses less samples overall (see Appendix D for the total number of samples used at each iteration)’. The number of samples used at each iteration increases but the algorithm requires much less iterations for higher values of the depth $h$. This results in a more sample efficient algorithm overall. Please see Fig. 3 (p. 28) and appendix D.2 (p. 27, l. 926-932) for a discussion along these lines.
>
> **The meaning of the dotted lines should be clarified in Figure 1 (Left).**
>
> The meaning of the dotted lines is given in the text in l. 317-318 in section 7: ‘(a) in dotted lines in Fig. 1 (left), $\eta_k$  equal to its lower bound in sec. 4, with the choice $c_k := \gamma^{2h(k+1)})$  (note the dependence on $h$); and (b) in solid lines, $\eta_k$  identical stepsize schedule across all values of $h$ with $c_k := \gamma^{2(k+1)}$ to isolate the effect of the lookahead.’ We will add this to the caption of Figure 1 for clarity.
>
> **Based on the theoretical result of the paper, does that mean larger $h$ leads to better performance? For Figure 1, what if we have a larger $h$?**
>
> We prove that a larger lookahead depth results in a better sample complexity and a faster suboptimality gap convergence rate. This improvement comes at the cost of an additional computational effort (compared to $h=1$ for 1-step policy improvement) to compute the lookahead value function at each iteration. However, our experiments suggest that the benefits of the faster convergence rate greatly outweigh the extra cost of computing the lookahead, in terms of both overall running time until convergence and sample complexity (in the inexact case). See section 7 and Appendix D for evidence and additional experiments.
>
> We have performed additional experiments with larger $h$ ($h=100$), see Figure 1 in the attached pdf. In this case, the algorithm converges in a single iteration. This is theoretically expected as computing the lookahead values with very large $h$ boils down to computing the optimal values like in value iteration with a large number of iterations.
>
> **The paper discusses some potential improvements in future work on the practical algorithm. However, they didn't discuss the limitations of the existing theoretical/practical results.**
>
> We thank the reviewer for the comment. We will further improve our discussion of the limitations of our work along the lines of investigating more general function approximation in even larger scale environments, adaptive lookahead depth selection as well as fully online estimators for the lookahead action values. We will also add further details regarding the computational tradeoff we mentioned for our lookahead algorithm.

---

> > ### Comment · Area_Chair_Wprb · 2024-08-13
> > **Reminder to address the author's rebuttal**
> >
> > Dear Pv8t, please try to respond to the author's feedback today and elaborate if your opinion changed in light of it.

---

### Author Rebuttal · Authors · 2024-08-07

We thank all the reviewers for their valuable time and feedback. We address the remaining concerns of the reviewers in our individual responses below. We have also performed additional experiments to support our responses. Please find figures relating to these experiments attached. These simulations include: a) testing the effect of a much larger lookahead depth (Reviewer Pv8t), b) comparing h-PI to h-PMD (Reviewers vPQx and SGix), c) testing h-PMD on two additional continuous control environments. Due to space constraints in our individual response to reviewer t6qC we include the rest of our response below in this general rebuttal. We hope that our responses address all of the questions of the reviewers, we also welcome any additional questions in the discussion period.

 --------

**End of response to reviewer t6qC**

**6. Can you provide more extensive empirical results across a variety of RL environments…?**

We have provided additional simulations in appendix D, p. 27-31, including tests in the `Catch’ environment from Deepmind’s bsuite and a grid world with variable size. As for robustness of h-PMD, we have investigated (in the appendix) the use of a different PMD regularizer (Figure 2) as well as varying  lookahead depths, chain lengths, grid world sizes, discount factors  (p. 30-31). That being said, please notice that our main contributions are theoretical and our simulations primarily serve an illustrative purpose.
During the rebuttal phase we performed additional experiments in continuous control tasks to illustrate the general applicability of our algorithm.

**7. How does h-PMD compare to other advanced RL algorithms like AlphaZero, in terms of performance and computational efficiency?..**

Our h-PMD algorithm can be linked to the class of AlphaZero algorithms (see l. 180-187). It has been shown in [13] that AlphaZero can be seen as a regularized policy optimization algorithm, drawing a connection to the standard PMD algorithm (with h = 1). We argue that AlphaZero is even more naturally connected to our h-PMD algorithm with lookahead values. We have also implemented a version of our h-PMD algorithm with Deep Mind’s MCTS implementation (see section D.5 for details). Conducting further experiments to compare our algorithm to AlphaZero on similar large scale settings would be interesting and would deserve its own separate study. Note that we are not aware of any theoretical convergence guarantee for AlphaZero which relies on many heuristics. In contrast, our $h$-PMD algorithm enjoys nice theoretical guarantees ranging from the exact to the inexact, stochastic and function approximation settings.

**8. What are the scalability limits of h-PMD in terms of state and action space sizes?...**

We prove in theorem 5.4 p. 7 that the sample complexity of h-PMD scales with the product of the state and action space sizes in the tabular setting. Then, we employ linear function approximation to show a value function gap performance bound that only scales with the dimension of the state action feature map without any dependence on the state action space size. Please see the response to reviewer QSY1 for a more detailed discussion.
Tree search methods (used as subroutine in h-PMD for lookahead value estimation) have been successfully used in practice in a number of works, including in the most successful applications of RL such as AlphaGo, AlphaZero and MuZero to name a few. Notice also that we also rely on Deep Mind’s efficient MCTS implementation in our experiments in appendix D.5. Hence, we do not foresee any feasibility issue of h-PMD given all the existing efficient tree search implementations that can be readily used as subroutines. As PG methods are also notoriously suitable for high-dimensional settings, our h-PMD also inherits such a potential. Our additional experiments show that our algorithm can be modified to work smoothly in large or even continuous state spaces (see Figure 3 in the rebuttal pdf), and is still reasonable when the lookahead depth is scaled up dramatically (see Figure 1). We will add to our paper a more detailed discussion along these lines.

**9. Can you elaborate on the trade-offs involved in the inexact setting of h-PMD? …**

Comparing Theorem 4.1 (exact setting) and Theorem 5.3 (inexact setting), there is an additive bias term in the value function bound due to inexact approximation of the lookahead value functions. In Theorem 5.4, we control this bias and make it arbitrarily small by choosing an adequate mini-batch size (of the order of $1/\epsilon^2$ where $\epsilon$ is the desired accuracy) for our Monte-Carlo lookahead estimator. We then derive the overall improved sample complexity of Theorem 5.4.
Hence the convergence rate (in terms of number of iterations) is now still geometric but up to the aforementioned bias and the overall sample complexity to reach an approximate optimal value function is improved. This is at the cost of computing approximate lookahead action values as discussed.

**10. Have you applied h-PMD to any real-world RL problems or industrial applications? …**

Thank you for this question which is definitely important. To support our contributions which are mainly theoretical, we have performed simulations in standard widely used RL benchmarks in the literature beyond existing simulations for h-PI which were restricted to the simple grid world environment (see e.g. [9]). These initial experiments are promising and we believe it will be interesting to test our algorithm on real-world RL problems.

**Some theoretical results assume exact policy evaluation, which may not be feasible …**

We completely relax exact policy evaluation in section 5. We do not suppose access to multi-step greedy policies, we rather provide a procedure to compute them approximately: See section 5.1 for lookahead Q-function estimation and its use in $h$-PMD in Eq. (5) as well as its extension to the function approximation setting (see Eq. (7) and section C).

---

### Decision · Program_Chairs · 2024-09-25

**Decision:**

Accept (poster)

**Comment:**

This work studies a new generalization of Policy Mirror Descent (PMD), h-PMD, which uses an h-step lookahead improvement for evaluating the Q function, instead of merely estimating the standard Q-function. The authors generalize the PMD analysis to establish the convergence of h-PMD and study its performance in the approximate setting, where the Q functions are approximately correct. They also present experimental results of h-PMD and investigate its performance as the lookahead parameter increases.
To my knowledge, this is the first work that formalizes the idea of combining mirror descent (or gradient descent) with h-step optimal Q functions. This formalization is useful, especially due to the fact that planning-type algorithms (e.g., MCTS) combined with RL-type techniques are known to be used in practice.

Technically speaking, the theoretical results are extensions of existing PMD analysis, with a careful application of the gamma^h contraction property of the Bellman operator. Further, the authors derived new for the h-PMD performance in the approximate.

Reviewers emphasized the empirical section of this work should be improved. Specifically:
1. The algorithm was tested only on a toy environment. In the rebuttal the authors shared results on additional environments.
2. The algorithm implementation in practice is not clear. h-PMD is only the high-level framework of the design, but the practical implementation is more nuanced. I would suggest the authors include the full algorithm in the appendix (e.g., how the loss functions are calculated per each batch? how often the h-step lookahead policy is calculated? are there any useful tricks worth mentioning?). I believe this can be incorporated in the camera ready version.
3. Is it worth scaling h to larger values in practice? Although the authors studied some aspects of this question in Appendix D, I believe further empirical investigation into this topic would be useful and should be presented in the main paper.

Even though there is more work to be done in this direction (e.g., how should we set h in practice?), I believe this work makes a substantial contribution and may pave the way for future advancements. I encourage the authors to incorporate the reviewers' and the aforementioned edits into the camera-ready version.